# DNOD: Deformable Neural Operators for Object Detection in SAR Images

**GVS Mothish**                                                    *mothishg@iisc.ac.in*
*Department of Computational and Data Sciences*
*Indian Institute of Science, Bengaluru, India.*

**J Rishi**                                                            *rishij@iisc.ac.in*
*Department of Computational and Data Sciences*
*Indian Institute of Science, Bengaluru, India.*

**Shobhit Kumar Shukla**                              *shobhitshukla6535@gmail.com*
*Department of Computational and Data Sciences*
*Indian Institute of Science, Bengaluru, India.*

**Deepak N. Subramani**                                      *deepakns@iisc.ac.in*
*Department of Computational and Data Sciences*
*Indian Institute of Science, Bengaluru, India.*

**Reviewed on OpenReview:** *https://openreview.net/forum?id=tjBqPJdQ72*

## Abstract

We introduce a deep neural operator framework aimed at object detection in remotely sensed Synthetic Aperture Radar (SAR) images. Recent research highlights the impressive performance of the End-to-End Object Detection Transformer (DETR). Nonetheless, in domains like SAR imaging, managing challenges such as speckle noise and the detection of small objects continues to be problematic. To address SAR object detection issues, we present the Deformable Neural Operator-Based Object Detection (DNOD) framework, tailored for SAR tasks. We develop two neural operators: Multi-Scale Fourier Mixing (MSFM) for the encoder and Multi-scale, multi-input Adaptive Deformable Fourier Neural Operator (MADFNO) for the decoder. Detailed evaluations and ablation studies show that DNOD exceeds existing methods, delivering significantly better results with an improvement of **+2.23** mAP on the SARDet-100k dataset, the largest SAR object detection compilation. The code is available at `https://github.com/quest-lab-iisc/DNOD`.

## 1 Introduction

Neural operators, emerging from computational physics, have demonstrated significant success in solving Partial Differential Equations (Kovachki et al., 2023). Rooted in operator theory, these neural operators learn mappings between function spaces of infinite dimensions, achieving notable success in numerous applications while inherently maintaining discretization invariance. Neural operators comprise three fundamental parts: (1) a lifting module, (2) an iterative kernel integral module, and (3) a projection module. Kernel integrals are operations within the spatial domain that ascertain global interdependencies crucial for learning infinite-dimensional function maps. Based on different forms of kernel integral computation, different neural operators such as Fourier Neural Operator (Li et al., 2020c), Graph Neural Operator (Li et al., 2020d), and Adaptive Fourier Neural Operator (Guibas et al., 2021) have been proposed. In a specific context, the attention mechanism utilized within transformers can be seen as a special case of kernel integral operations (Kovachki et al., 2023). Recently, neural operators have demonstrated superior performance in computer vision applications such as super-resolution (Wei & Zhang, 2023; Liu & Tang, 2025), and inpainting (Guibas

et al., 2021). However, neural operators have not been employed for the task of object detection in Synthetic Aperture Radar (SAR) imagery, a gap this paper addresses.

SAR is an advanced active microwave sensing technology capable of acquiring high-resolution images regardless of weather conditions, illumination, or time of day (Tirandaz et al., 2020; Brown, 1967; Moreira et al., 2013). SAR images can provide much more useful information and be effective in military reconnaissance, marine surveillance, port management, and disaster response applications (Guan et al., 2023; Zhang et al., 2022a; Chen et al., 2020; Zhang et al., 2022b). As modern satellites provide increasingly accessible high-resolution, large-scale SAR images, the demand for sophisticated methods to effectively process large data volumes has increased. Consequently, the precise detection of targets from complex terrestrial environments using SAR images is of great practical importance (Sharifzadeh et al., 2019).

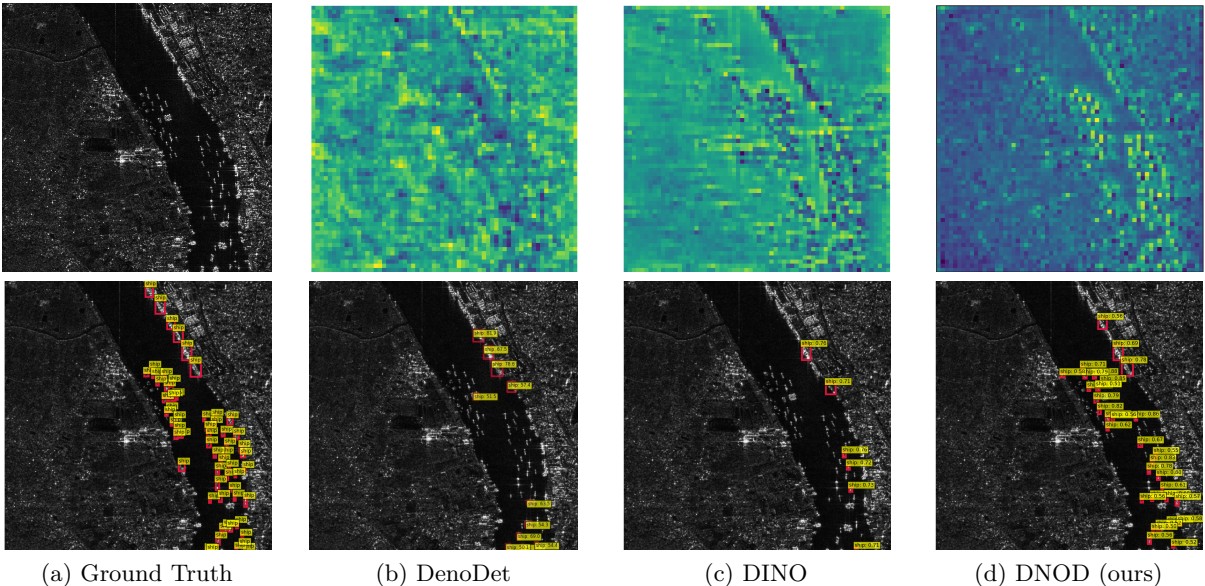

    (a) Ground Truth         (b) DenoDet         (c) DINO         (d) DNOD (ours)

Figure 1: **Qualitative evaluation of DNOD for SAR image object detection against recent state-of-the-art models**: **(a)** Top row is the input SAR image and bottom row is the ground truth of bounding boxes. **(b)-(d)** The top row shows the learned representation by encoder operator, and bottom row shows detected objects by DenoDet, DINO and our DNOD. Notably, DNOD encoder operator (MSFM) effectively differentiates small objects and other features better than DenoDet (TransDeno) and DINO (Deformable), underscoring the neural operator's efficiency and superior ability to detect small objects compared to other leading models.

Numerous SAR object detection methods have been proposed, from traditional methods (Nitzberg, 2007; Migliaccio et al., 2012) to CNN-based methods (Gao et al., 2021). In recent developments, transformers have been introduced for object detection, explicitly known as DETR (Detection Transformers; Carion et al. (2020)), and have shown superior performance compared to traditional hand-crafted feature engineering methods. Various iterations and modifications of DETR, such as Deformable DETR and DAB DETR, have exhibited outstanding results in the field of object detection, further enhancing their effectiveness and application (Zhu et al., 2021; Liu et al., 2022a; Zhao et al., 2024; Zhang et al., 2023a; Lin et al., 2023b; Meng et al., 2021). An extensive discussion of the different methodologies for object detection is available in the Appendix A.1. Even with the advent and introduction of multiple variants of the DETR model, its effectiveness in SAR images has been less than satisfactory (Dai et al., 2024). There have been different challenges associated with SAR images, specifically *(i)* speckle noise interference (Yue et al., 2020); and *(ii)* small target challenges (Wan et al., 2021). Given the DETR framework's notable success in object detection, attributed to its foundation on the transformer architecture, it is feasible to integrate neural operators into the DETR framework specifically for executing SAR object detection tasks.

This paper presents the Deformable Neural Operator for object Detection (DNOD) in SAR images. DNOD is trained and evaluated on the COCO-level large-scale multi-class SAR object detection dataset, SARDet-100k (Li et al., 2024b). For an illustrative example of the diversity of the dataset, refer to Appendix A.3. Our methodology employs neural operator architecture within the framework of End-to-End Object Detection using transformers (DETR). We introduce two architectural components drawn from neural operator concepts: *(i)* The Multi-Scale Fourier Mixing (MSFM) Encoder and *(ii)* The Multi-Scale Adaptive Deformable Fourier Neural Operator (MADFNO) Decoder. There are two main advantages of using neural operators for SAR object detection: *(i)* Fourier component in the neural operator reduces the effect of speckle noise in SAR images; and *(ii)* the discretization invariance property of the neural operator reduces the challenges related to small target detection. Section 4.4 describes comprehensive visual insights.

In summary, our main contributions are as follows.

1. To the best of our knowledge, this is the first work to introduce neural operators for object detection applications.

2. We develop two novel architectural components, MSFM and MADFNO, specifically designed to enhance object detection performance in SAR imagery. Visualizations of the learned representations highlight the capability of our new operators (Figure 1).

3. We integrate our proposed neural operators within the DETR framework to achieve effective SAR object detection.

4. Through comprehensive empirical evaluation, we demonstrate that our method achieves SoTA performance for SAR object detection compared to existing object detection techniques.

## 2 Preliminaries: Neural Operator

Neural operators (Kovachki et al., 2023) are built to learn function-to-function mappings. Initially introduced to solve PDEs, they have recently been applied to computer vision tasks. Consider an operator $\mathcal{G} \colon \mathcal{A} \to \mathcal{U}$ that acts between the function spaces $\mathcal{A}$ and $\mathcal{U}$. Neural operators are the parametric map $\mathcal{G}_\phi \colon \mathcal{A} \to \mathcal{U}$ that approximates $\mathcal{G}$ and is learned from empirical data or physical principles. Formally, the parametrized neural operator can be expressed as

$$\mathcal{G}_\phi := \mathcal{Q} \circ \sigma(\mathcal{W}_T + \mathcal{K}_T + b_T) \circ \cdots \circ \sigma(\mathcal{W}_1 + \mathcal{K}_1 + b_1) \circ \mathcal{P}, \tag{1}$$

where, $\mathcal{P}$ and $\mathcal{Q}$ serve as the lifting and projection operators. The lifting operator raises the codomain to a higher-dimensional representation space, while the projection operator reduces the codomain to the output dimension. These operators are typically parameterized as multilayer perceptrons and act point-wise on functions. The function $\sigma$ represents pointwise nonlinearity. Each layer $t = 1, ..., T$ includes a local operator $\mathcal{W}_t$ (usually parameterized by a point-wise neural network), a kernel integral operator $\mathcal{K}_t$, and a bias function $b_t$. Given an intermediate functional representation $v_t$ with domain $D$ in the $t$-th hidden layer, a kernel integral operator $\mathcal{K}_\phi$ is defined as

$$(\mathcal{K}_\phi v_t)(x) := \int_D \kappa_\phi(x, y, v_t(x), v_t(y)) v_t(y) \, \mathrm{dy}, \tag{2}$$

where the kernel $\kappa_\phi$ is a learnable neural network with parameter $\phi$. Different neural operators (Equation 1) are defined on the basis of their kernel integrals (Equation 2). Each of these operator layers is expressed as $\{v_t \colon D_t \to \mathbb{R}^{d_{v_t}}\} \mapsto \{v_{t+1} \colon D_{t+1} \to \mathbb{R}^{d_{v_{t+1}}}\}$ using

$$v_{t+1}(x) = \sigma_{t+1} \left( W_t v_t(x) + \int_{D_t} \left( \kappa^{(t)}(x, y) v_t(y) \right) dv_t(y) + b_t(x) \right) \ \forall x \in D_{t+1}. \tag{3}$$

# 3 Methodology

Building on top of the DETR framework (see Appendix A.2; Carion et al. (2020)), we develop our DNOD model (Figure 2) by introducing two new neural operator architectural components: *(i)* The Multi-Scale Fourier Mixing (MSFM) Encoder (Sect. 3.1) and *(ii)* The Multi-Scale Adaptive Deformable Fourier Neural Operator (MADFNO) Decoder (Sect. 3.2). Our proposed neural operator modules are specifically designed to learn multi-scale feature maps that have been shown to benefit modern object detection frameworks (Liu et al., 2020; Zhu et al., 2021).

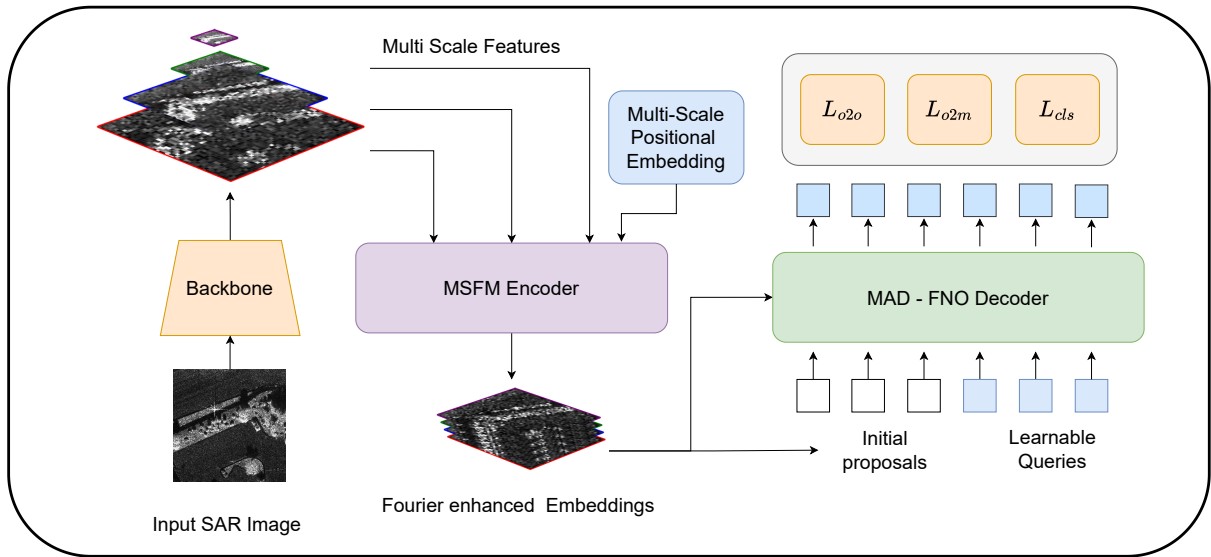

Figure 2: **Overview of the DNOD framework:** DNOD architecture processes input SAR images using a backbone and MSFM encoder to extract multi-scale, Fourier-enhanced embeddings. These features are passed to the MADFNO decoder along with initial proposals and learnable queries. The decoder outputs are supervised by three loss functions (i) classification (ii) one to one matching loss and (iii) one to many matching, to guide robust SAR image detection.

## 3.1 Multi-Scale Fourier Mixing (MSFM Encoder)

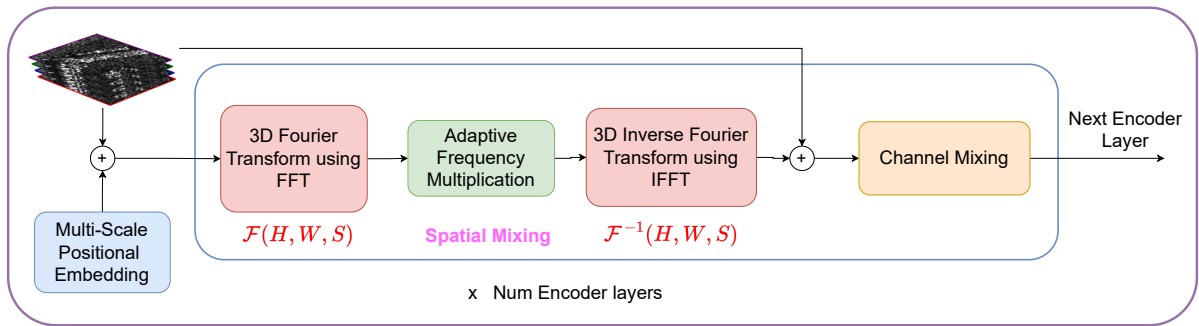

Figure 3: **MSFM Encoder:** Multi-scale features combined with positional embedding are fed into the encoder. Initially, a Fourier transform is executed across scale, height, and width. This is succeeded by spatial mixing and then an inverse transform. Subsequently, channel mixing is applied, and the resulting output is passed on to the succeeding encoder layer. This entire sequence is repeated for the specified number of encoder layers, ultimately yielding Fourier-enhanced embeddings.

Within the DETR framework, multiple encoders have been utilized, such as Vision Transformer (Carion et al., 2020) and Deformable Transformer (Zhu et al., 2021). However, these encoders work in the image domain, thereby failing to segregate coherent speckle noise in SAR images that are intermixed with features (Dai et al., 2024). Removal of noise before detection can result in missing crucial details for subsequent tasks, rendering it an ill-posed problem (Sun et al., 2022), thus necessitating an architecture adept at handling noisy features. We introduce a neural operator framework, named Multi-Scale Fourier Mixing (MSFM), adept at effectively handling multi-scale features and speckle denoising in the frequency domain (Figure 3). Our MSFM is motivated by the success of the spectral convolutions used in the Fourier Neural Operator (FNO) (Li et al., 2020c) and the efficient token mixer introduced in the Adaptive Fourier Neural Operator (AFNO) (Guibas et al., 2021). These operators employ the convolution theorem to transform convolutions in the spatial realm to element-wise multiplications with block diagonal structure in the Fourier domain. The main distinction between AFNO and our MSFM is that MSFM is specifically designed to manage multi-scale features in the Fourier domain, essential for tasks like denoising and object detection.

The MSFM kernel integral for a continuous multi-scale variable $X \in D$ with a kernel function $\kappa$ at a specific token $s$ can be expressed as

$$\mathcal{K}(X)(s) = \mathcal{F}^{-1}\left(\mathcal{F}(\kappa) \cdot \mathcal{F}(X)\right)(s) \quad \forall s \in D, \tag{4}$$

where $\cdot$ denotes matrix multiplication, and $\mathcal{F}, \mathcal{F}^{-1}$ denotes the continuous Fourier transform and its inverse.

In practice, each MSFM encoder block begins with spatial mixing across multiple scales via the Fourier transform ($z_{m,n}$ where (m,n) is the index per token), which is followed by channel mixing ($\tilde{z}_{m,n}^{(l)}$) with a block diagonal structure and the inverse Fourier transform ($y_{m,n}$). The final Fourier-enhanced embeddings (Figure 2) are obtained after multiple encoder blocks. Mathematically, each encoder block can be expressed as

$$z_{m,n} = [FFT(X)]_{m,n}, \quad \tilde{z}_{m,n}^{(l)} = W_{m,n}^{(l)} z_{m,n}^{(l)}, \quad l = 1, \ldots, p, \quad y_{m,n} = [IFFT(SoftShrink(\tilde{z}_{m,n}))]. \tag{5}$$

The above formulation improves efficiency, generalization, and speckle noise removal through block-diagonal channel mixing, shared MLP weights, and soft-thresholding. The pseudo code for the MSFM encoder is provided in Appendix A.5.

## 3.2 Multi-Scale Adaptive Deformable Fourier Neural Operator (MADFNO Decoder)

DETRs face challenges in settings with limited feature resolution, resulting in below-average performance in detecting objects across varying scales. This constraint impairs the model's ability to detect smaller objects prevalent in SAR imagery. Furthermore, employing a transformer decoder hinders the convergence speed. To mitigate these issues, deformable attention modules (Zhu et al., 2021) have been implemented in generic object detection. However, as mentioned earlier (Section 3.1), successful SAR object detection demands addressing both speckle noise and multi-scale features, calling for a neural operator. Moreover, in DETRs, the decoder's task is to query objects based on the features produced by the encoder. This requires a neural operator that can handle multiple inputs. Recent studies (Jiang et al., 2024; Lehmann et al., 2025) have explored multi-input neural operators; however, none incorporate deformable methods that are essential for multi-scale feature extraction.

Thus, we introduce the MADFNO (Figure 4), a novel neural operator designed for handling multiple inputs, multi-scale scenarios, and incorporating deformable technique with Fourier mixing.

MADFNO takes object queries ($Q$) as input along with features of the encoder ($E_o$). First, object queries undergo self-Fourier mixing ($\mathcal{M}$) with a kernel function $\mu$ at a specific token $s$, defined as

$$\mathcal{M}(Q)(s) = \mathcal{F}^{-1}\left(\mathcal{F}(\mu) \cdot \mathcal{F}(Q)\right)(s) \quad \forall s \in D. \tag{6}$$

Subsequently, these object queries are combined with encoder embeddings ($E_o$) through the deformable operator ($D$) to obtain the refined object queries, $D(\mathcal{M}(Q), E_o)$, evaluated as

$$D(Z, E_o)(s) = [B(T_k, E_{ok}), \cdots B(T_1, E_{o1})]; \text{where} \quad Z = \mathcal{M}(Q). \tag{7}$$

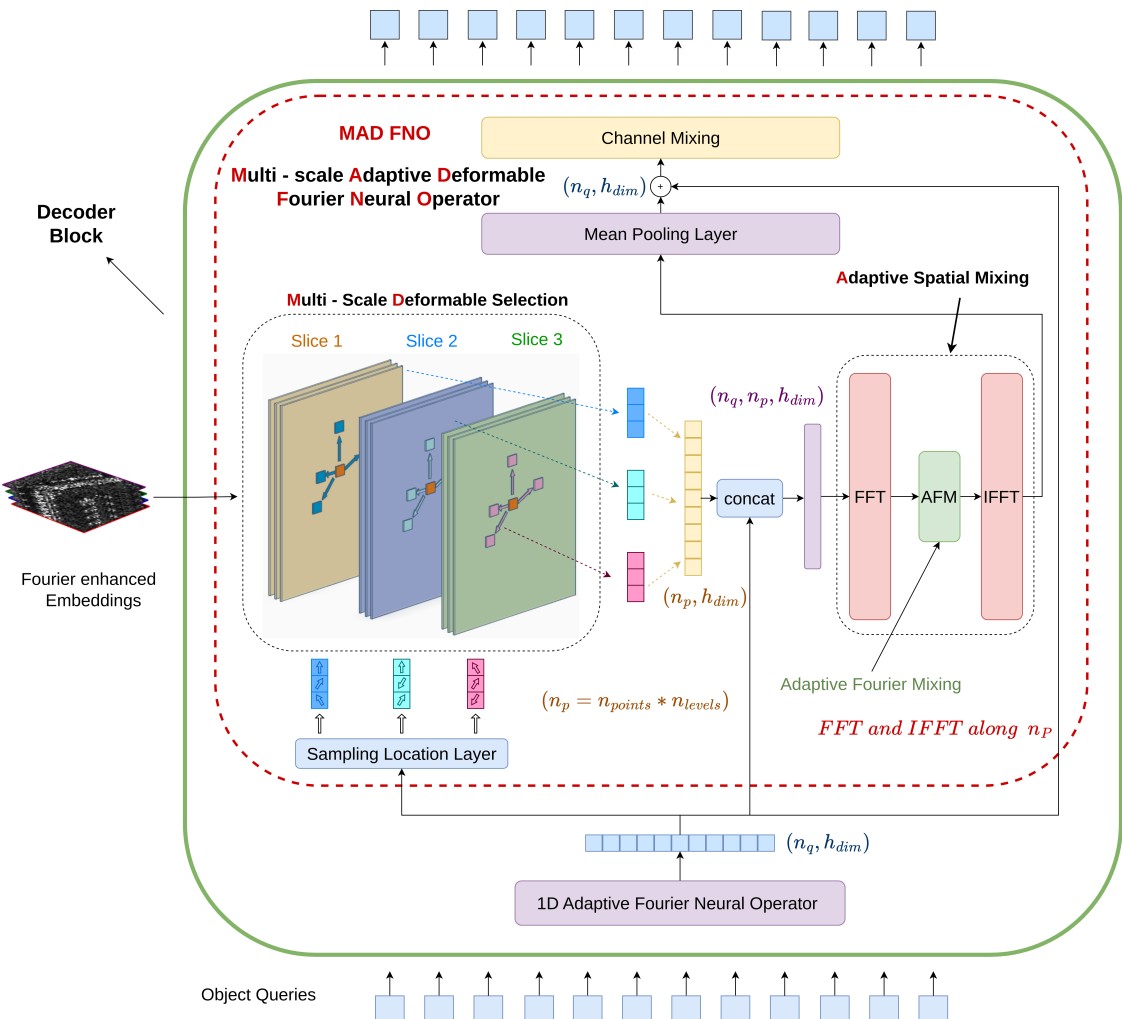

Figure 4: **MADFNO Decoder:** Object queries along with Fourier-enhanced encoder embeddings serve as input to the decoder. The deformable operator extracts features from the encoder embeddings through sampling locations determined by the object queries. These extracted features are subsequently subjected to Fourier mixing, resulting in the final object queries for the subsequent decoder layer.

**Deformable operator (D):** Rather than selecting features directly from the encoder output, we partition the encoder embeddings $(E_o)$ into $k$ different slices such that $E_o = \cup_{i=1}^{k} E_{oi}$ and $\cap_{i=1}^{k} E_{oi} = \phi$, which implies that $E_o = [E_{o1}, E_{o2}, \cdots, E_{ok}]$, where [,] denotes the concatenation operator. This sliced sampling facilitates feature selection by concentrating each slice on distinct relations within $E_o$, similar to the multi-head attention mechanism in traditional transformer models. Note that each slice contains multi-scale features as encoder embeddings.

The sampling locations $(T)$ necessary for the deformable operator are obtained from the initial reference points $(R_p)$, which are in turn obtained from encoder embeddings $(E_o)$ and sampling residuals $(r)$, such that $T = R_p + r$ where $r$ is learnable and $R_p$ is estimated from $E_o$. The sampling residuals are learned via the sampling location layer $(SL)$, that is,

$$r_1, r_2, \cdots r_k(s) = SL(Z)(s) \quad where \quad Z = \mathcal{M}(Q), \tag{8}$$

where $r_j$ represents the sampling residuals for the $j^{th}$ slice. To sample features from each slice, a sampling location layer takes object queries $M(Q)$ as input and outputs sampling residuals per slice, which are further added to reference points per slice $R_{pj}$ to obtain final sampling locations per slice $T_j$, i.e., $T_j = R_{pj} + r_j$.

The final sampling locations derived are continuous; consequently, bilinear interpolation is used to extract features from the encoder embeddings, denoted as $B(T_j, E_{oj})$ for the $j^{th}$ slice. All these slices are then concatenated into a single slice of sampled encoder embeddings with dimensions $(n_p, h_{dim})$, where $n_p = n_{points} * n_{scale}$, with $n_{points}$ as a hyperparameter denoting the required number of sampling points per feature scale. These embeddings are then concatenated with object queries to form combined embeddings per object query with dimensions $(n_q, n_p, h_{dim})$. Next, a Fourier transform is performed across these sampled features $(n_p)$, followed by spatial mixing and an inverse Fourier transform leading to Fourier mixing of queries with sampled encoder embeddings. This process is followed by a mean pooling operation along the selected features and subsequent channel mixing to produce the final refined object queries. Overall, the kernel integral of the multi-input neural operator MADFNO can be expressed as

$$\mathcal{K}(Q, E_o)(s) = \mathcal{M} \circ F^{-1}\left(F(\kappa) \cdot F([Z, D(Z, E_o)])\right)(s) \quad \forall s \in D, \quad Z = \mathcal{M}(Q). \tag{9}$$

## 4 Experiments

We first present the datasets used for our experimental analysis and the implementation procedure. We then proceed to evaluate the performance of DNOD in comparison to baseline models. Finally, we performed an ablation study to illustrate the significance of each component and its impact on overall effectiveness.

### 4.1 Experimental Setup

#### 4.1.1 Datasets

The SARDet-100k dataset is used in our experiments focused on object detection. This dataset comprises 116,598 images and 245,653 instances classified into six categories: Aircraft, Ship, Car, Bridge, Tank and Harbor. As the first extensive SAR object detection dataset, SARDet-100K is similar in scale to the widely recognized COCO dataset (118K images) (Lin et al., 2014), a benchmark for general object detection. SARDet-100k is constructed by integrating nine different datasets focused on SAR object detection. These datasets exhibit varied polarities and encompass a wide range of resolutions, ranging from 0.1 to 25 meters. The data is collected by utilizing six different satellites, each operating within four diverse frequency bands. The extensive scope and diversity of the SARDet-100K dataset, as outlined in Table 1, accurately represent the real-world obstacles encountered in deploying SAR object detection models across different data sources. We employ intensity images in the present work.

Table 1: Source datasets in SARDet-100K (Li et al., 2024b). Target categories S: ship, A: aircraft, C: car, B: bridge, H: harbour, T: tank.

| Datasets | Target | Res. (m) | Band | Polarization | Satellites | License |
|---|---|---|---|---|---|---|
| AIR_SARShip (Xian et al., 2019) | S | 1.3m | C | VV | GF-3 | - |
| HRSID (Wei et al., 2020) | S | 0.5~3m | C/X | HH, HV, VH, VV | S-1B, TerraSAR-X, TanDEM-X | GNU General Public |
| MSAR (Xia et al., 2022) | A, T, B, S | ≤ 1m | C | HH, HV, VH, VV | HISEA-1 | CC BY-NC 4.0 |
| SADD (Zhang et al., 2022c) | S | 0.5~3m | X | HH | TerraSAR-X | - |
| SAR-AIRcraft (Zhirui et al., 2023) | A | 1m | C | Uni-polar | GF-3 | CC BY-NC 4.0 |
| ShipDataset (Wang et al., 2019) | S | 3~25m | C | HH, VV, VH, HV | S-1, GF-3 | - |
| SSDD (Zhang et al., 2021b) | S | 1~15m | C/X | HH, VV, VH, HV | S-1, RadarSat-2, TerraSAR-X | Apache 2.0 |
| OGSOD (Wang et al., 2023) | B, H, T | 3m | C | VV/VH | GF-3 | - |
| SIVED (Lin et al., 2023a) | C | 0.1, 0.3m | Ka, Ku, X | VV/HH | Airborne SAR synthetic slice | - |

#### 4.1.2 Implementation Details

In DNOD and other baselines, the same ResNet-50 backbone (pre-trained on the ImageNet-1K dataset) is utilized for fair comparison of object detection models. DNOD contains MSFM encoder and MADFNO decoder each having 3 layers, utilizing a hidden feature dimension of 256. With 1200 decoder object queries, training is accomplished through both one-to-one (Zhang et al., 2023a) and one-to-many matching (Zhao et al., 2024) losses. Based on Zhang et al. (2023a), we use L1 and GIoU losses for the regression of the

bounding box and adopt focal loss with $\alpha = 0.25$ and $\gamma = 2$ for classification. Additionally, techniques such as Look Forward Twice and Mixed Query Selection are integrated (Zhang et al., 2023a). Following the DETR framework, auxiliary losses are introduced after each decoder layer. The model underwent 56 epochs of training on 2 Nvidia RTX A6000 GPUs, with a cumulative batch size of 16. Initially, the learning rate was configured at $1 \times 10^{-4}$, which was reduced by a factor of 0.1 after 52 epochs. We utilized the AdamW optimizer, with a weight decay rate of $1 \times 10^{-4}$. Further implementation details are in the Appendix A.6.

## 4.2 Results

Table 2: Comparison of SAR Object detection methods on the **SARDet-100k** dataset. Bold indicates it is better than all models, and an underline is the second best. All the models used a ResNet-50 backbone pretrained (Pre.) on ImageNet. Baseline results are from (Dai et al., 2024)

| Method | Pre. | FLOPs | #Params | **mAP** | AP@50 | AP@75 | $AP_S$ | $AP_M$ | $AP_L$ |
|---|---|---|---|---|---|---|---|---|---|
| *One-stage* | | | | | | | | | |
| FCOS | IN | 51.57G | 32.13M | 52.52 | 85.82 | 54.93 | 47.01 | 66.13 | 57.82 |
| GFL | IN | 52.36G | 32.27M | 55.01 | 85.16 | 58.87 | 49.44 | 67.29 | 60.45 |
| RepPoints | IN | 48.49G | 36.82M | 51.66 | 86.43 | 53.99 | 46.66 | 63.26 | 53.78 |
| ATSS | IN | 51.57G | 32.13M | 54.95 | 87.60 | 58.25 | 49.89 | 67.94 | 58.97 |
| CenterNet | IN | 51.55G | 32.12M | 53.91 | 86.17 | 57.31 | 48.88 | 66.22 | 57.74 |
| PAA | IN | 51.57G | 32.13M | 52.20 | 85.71 | 54.80 | 46.00 | 63.90 | 57.61 |
| PVT-T | IN | 42.19G | 21.43M | 46.10 | 77.55 | 49.00 | 38.01 | 59.53 | 53.35 |
| RetinaNet | IN | 52.77G | 36.43M | 46.48 | 77.74 | 48.94 | 40.25 | 59.35 | 50.26 |
| TOOD | IN | 50.52G | 30.03M | 54.65 | 86.88 | 58.41 | 50.20 | 66.72 | 58.60 |
| DDOD | IN | 45.58G | 32.21M | 54.02 | 86.64 | 57.23 | 49.33 | 64.70 | 58.02 |
| VFNet | IN | 48.38G | 32.72M | 53.01 | 84.32 | 56.32 | 47.37 | 65.39 | 57.99 |
| AutoAssign | IN | 51.83G | 36.26M | 53.95 | 89.58 | 55.96 | 50.14 | 63.40 | 54.73 |
| YOLOF | IN | 26.32G | 42.46M | 42.83 | 74.95 | 43.18 | 33.73 | 56.19 | 53.57 |
| YOLOX | IN | 8.53G | 8.94M | 34.08 | 66.77 | 31.31 | 28.49 | 43.06 | 28.95 |
| *Two-stage* | | | | | | | | | |
| Faster R-CNN | IN | 63.2G | 41.37M | 39.22 | 70.04 | 39.87 | 32.55 | 47.23 | 42.02 |
| Cascade R-CNN | IN | 90.99G | 69.17M | 53.55 | 87.33 | 56.81 | 49.09 | 62.89 | 48.68 |
| Dynamic R-CNN | IN | 63.2G | 41.37M | 49.75 | 80.96 | 53.91 | 43.12 | 59.72 | 54.77 |
| Grid R-CNN | IN | 0.18T | 64.47M | 50.05 | 80.58 | 53.49 | 42.43 | 62.01 | 52.70 |
| Libra R-CNN | IN | 64.02G | 41.64M | 52.09 | 83.54 | 55.81 | 45.85 | 63.52 | 55.40 |
| ConvNeXt | IN | 63.84G | 45.07M | 53.15 | 85.52 | 57.28 | 45.67 | 64.55 | 58.61 |
| ConvNeXtV2 | IN | 0.12T | 0.11G | 53.91 | 86.01 | 58.90 | 47.63 | 64.67 | 59.57 |
| LSKNet | IN | 53.73G | 30.99M | 52.39 | 85.07 | 56.96 | 45.15 | 63.59 | 59.16 |
| *End2End* | | | | | | | | | |
| DETR | IN | 24.94G | 41.56M | 45.73 | 78.57 | 46.87 | 37.01 | 58.16 | 55.58 |
| Deformable DETR | IN | 51.78G | 40.10M | 52.00 | 88.77 | 54.03 | 46.99 | 63.58 | 58.55 |
| DAB-DETR | IN | 28.94G | 43.70M | 43.31 | 78.14 | 43.10 | 34.82 | 56.34 | 52.62 |
| Conditional DETR | IN | 28.09G | 43.45M | 44.04 | 77.88 | 44.40 | 35.25 | 56.47 | 52.86 |
| DINO | IN | 81.41G | 46.67M | 53.40 | 87.82 | 56.15 | 47.05 | 66.19 | 61.98 |
| DenoDet | IN | 52.69G | 65.78M | 55.88 | 85.81 | 60.16 | 50.63 | 68.47 | 60.96 |
| **DNOD (ours)** | **IN** | 40.36G | 35.67M | **56.96** | **90.36** | 59.69 | **52.94** | **71.22** | **65.43** |
| **Promotion** | - | - | - | **+1.08** | **+0.78** | - | **+2.31** | **+2.75** | **+4.47** |
| **DNOD large (ours)** | **IN** | 78.01G | 43.10M | **58.11** | **91.44** | **61.31** | **55.31** | **70.30** | **64.36** |
| **Promotion** | - | - | - | **+2.23** | **+1.86** | **+1.15** | **+4.68** | **+1.83** | **+3.40** |

To assess the effectiveness of our model, we conducted a comparative analysis with 28 baseline methods (details in Appendix A.4), encompassing a variety of categories, including one-stage, two-stage, and end-to-end approaches. This selection, which includes convolution-based models, transformer-based models, and single-shot detectors such as YOLO, ensures a robust evaluation of our proposed model. The baseline results were obtained from DenoDet (Dai et al., 2024). We report average precision (AP) calculated using standard COCO (Lin et al., 2014) evaluation metrics. We report AP at different IOU thresholds and on different object scales, small ($AP_S$), medium ($AP_M$) and large ($AP_L$). We report our primary metric, COCO mAP, which calculates the mean of AP scores on 10 IoU thresholds from 0.50 to 0.95 with a step size of 0.05.

### 4.2.1 Quantitative Results

We have benchmarked DNOD with 28 diverse baselines (Table 2). Our DNOD achieved SoTA performance across all metrics at different IoU thresholds and on all object scales: small, medium, and large. DNOD uses 3 scales with $32 \times 32$ resolution scale derived from the backbone as the primary feature map. Compared to the previous SoTA model (DenoDet 4 Scale (Dai et al., 2024)) on SARDet-100k, our model demonstrated a **+1.08** mAP improvement, with a **45.7%** reduction in parameters and **23.4%** fewer GFLOPs. This computational efficiency is due to the neural operator-based encoder and decoder we introduced. We also developed a larger version of our model called 'DNOD Large' to further enhance the performance on SAR object detection. In this design, we utilized four scales with a $64 \times 64$ resolution scale derived from the backbone as primary feature maps. This enhancement led to an increase in mAP to **58.11**, marking an improvement of **+2.23** over the previous SoTA.

### 4.2.2 Qualitative Results

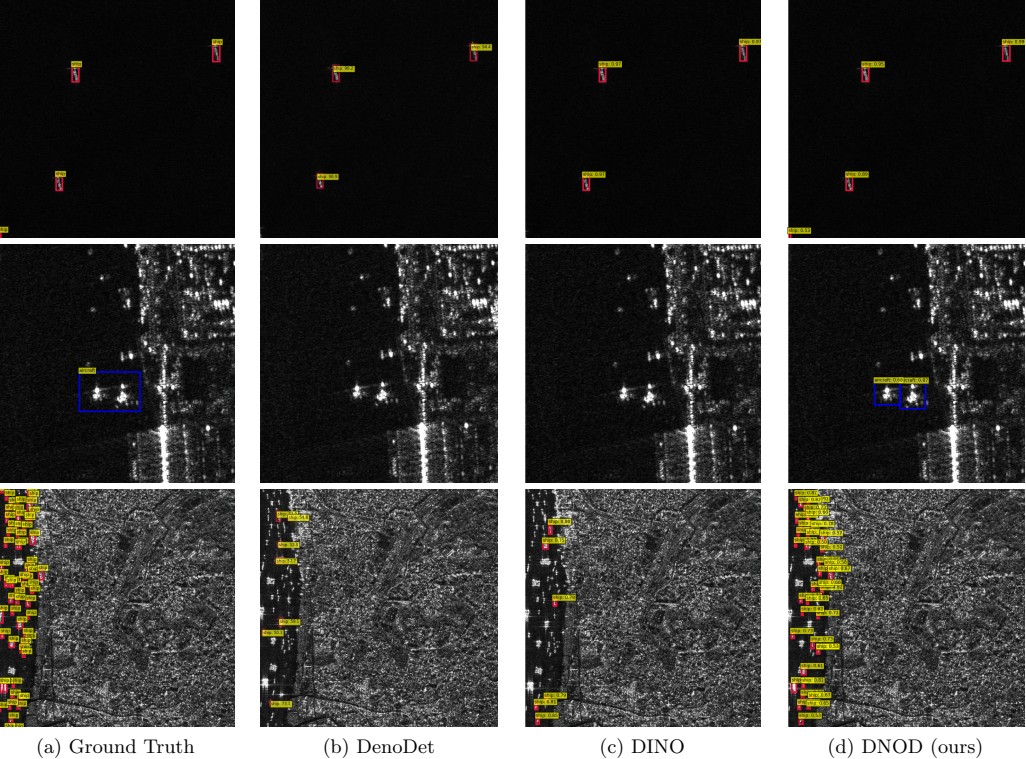

    (a) Ground Truth          (b) DenoDet          (c) DINO          (d) DNOD (ours)

Figure 5: **Qualitative Assessment**: **(Row 1):** DNOD effectively identified a partially visible object, which was not achieved by the other models. **(Row 2):** DNOD accurately detected an aircraft, which the other leading models failed to recognize. **(Row 3):** DNOD succeeded in detecting all ships, unlike other models. This analysis shows the superior ability of DNOD to detect objects in diverse settings.

We present the qualitative analysis of DNOD compared with two leading object detection models, (1) DenoDet (Dai et al., 2024), which is a leading model in SAR object detection, and (2) DINO (Zhang et al., 2023a), a leading model for generic object detection. This ensures a diverse evaluation of our model against both SAR object detection and generic object detection models. Figure 5 presents 4 different scenarios, (1) Partially occluded: DNOD successfully identified a partially occluded image, a feat not accomplished by the other models. (2) Similar objects in close proximity: DNOD precisely detected two different aircrafts that the other leading models had missed. (3) Smaller objects & (4) Crowded scenario of smaller objects: In both the third and fourth rows, DNOD was able to identify almost all of the ships which are much smaller objects compared to the image size, in contrast to the other leading counterparts. All predictions were assessed with a classification confidence greater than 0.5. This qualitative assessment underscores the superiority of DNOD in SAR object detection compared to other leading models, attributed to the discrete invariance property of neural operators (Kovachki et al., 2023).

### 4.3 Ablation Study

Table 3: Ablation comparison of (**MSFM**) encoder and (**MADFNO**) decoder

| Encoder | Decoder | **mAP** | $AP_{50}$ | $AP_{75}$ | $AP_S$ | $AP_M$ | $AP_L$ |
|---------|---------|---------|-----------|-----------|--------|--------|--------|
| Deformable | Deformable | 52.00 | 88.77 | 54.03 | 46.99 | 63.58 | 58.55 |
| **MSFM** | Deformable | 54.47 | 88.58 | 57.47 | 50.73 | 68.94 | 62.04 |
| Promotion | - | **+2.47** | - | **+3.44** | **+3.74** | **+5.36** | **+3.49** |
| MHSA | **MADFNO** | 55.80 | 89.67 | 58.51 | 51.78 | 69.86 | 63.42 |
| Promotion | - | **+3.80** | **+0.90** | **+4.48** | **+4.79** | **+6.28** | **+4.87** |
| **MSFM** | **MADFNO** | 56.96 | 90.36 | 59.69 | 52.94 | 71.22 | 65.43 |
| Promotion | - | **+4.96** | **+1.59** | **+5.66** | **+5.95** | **+7.64** | **+6.88** |

**Effect of MSFM and MADFNO:** To evaluate the effectiveness of the proposed architecture, we perform an ablation study comparing various encoder-decoder configurations. As shown in Table 3, replacing the conventional deformable encoder with our MSFM encoder consistently improves performance across all $AP$ metrics. Especially mean average precision (mAP) improved by **+2.47** with our encoder on SAR imagery. Furthermore, incorporating the MADFNO decoder in place of traditional deformable decoding significantly improves detection accuracy and has shown **+3.80** improvement in mAP. The combination of the MSFM encoder and the MADFNO decoder achieves the best performance, achieving an AP of **56.96**, a promotion of **+4.96**, with notable improvements in $AP_S$(52.94) (**+5.95**) and $AP_M$(71.22) (**+7.64**), demonstrating the effectiveness of the two neural operators.

Table 4: Ablation comparison of number of feature scales

| # Feature Scales | **mAP** | $AP_{50}$ | $AP_{75}$ | $AP_S$ | $AP_M$ | $AP_L$ |
|------------------|---------|-----------|-----------|--------|--------|--------|
| DenoDet 4 Scale (Previous SoTA) | 55.88 | 85.81 | 60.16 | 50.63 | 68.47 | 60.96 |
| DNOD 2 Scale | 55.56 | 89.75 | 57.66 | 51.36 | 70.63 | 63.49 |
| Promotion wrt SoTA | - | **+3.94** | - | **+0.73** | **+2.16** | **+2.53** |
| DNOD 3 Scale | 56.96 | 90.36 | 59.69 | 52.94 | 71.22 | 65.43 |
| Promotion wrt SoTA | **+1.08** | **+4.55** | - | **+2.31** | **+2.75** | **+4.47** |
| DNOD 4 Scale | 58.11 | 91.44 | 61.31 | 55.31 | 70.30 | 64.36 |
| Promotion wrt SoTA | **+2.23** | **+5.63** | **+1.15** | **+4.68** | **+1.83** | **+3.40** |

**Effect of Different Scales:** We examine the effect of varying the number of feature scales in our architecture, adjusting it from 2 to 4. Table 4 reveals that increasing the feature scales consistently improves the overall detection performance. Transitioning to three scales significantly boosts the average precision ($AP$) to **56.97** from **55.56** at two scales, and also improves $AP_{75}$ to **61.31** and $AP_S$ of **55.31**, highlighting the effectiveness of rich comprehensive multi-scale representations for detecting objects of differing sizes. These findings confirm the crucial role of integrating more scales in our framework.

We have also benchmarked DNOD with a ResNet-18 backbone, analyzed its robustness by varying scales and the number of queries, and provided a detailed computational analysis with varying batch sizes, including latency and energy consumption. Furthermore, a FLOPs vs. mAP comparison is included to assess DNOD's effectiveness against baselines, and experiments with the DNOD encoder using Non-Neural Operator (NO) spectral mixers are presented to justify the impact of MSFM. All these results are provided in Appendix A.7.

### 4.4 DNOD in Action: Visual Insights

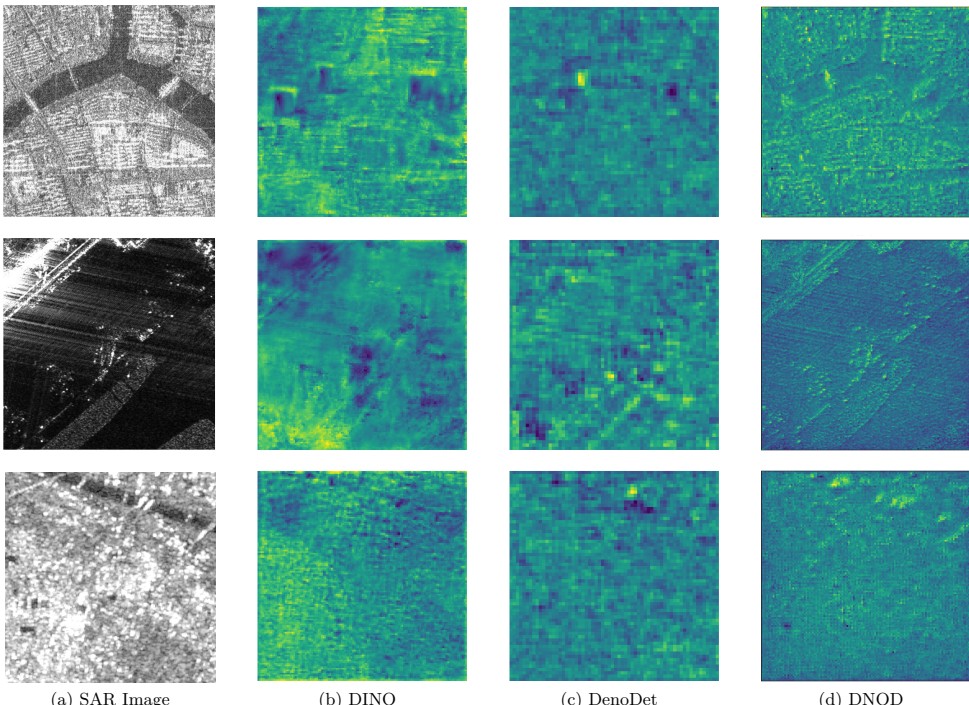

| (a) SAR Image | (b) DINO | (c) DenoDet | (d) DNOD |

Figure 6: **Comparative Evaluation of Encoder Outputs**: DNOD vs. Other leading Models. For images impacted by speckle noise (rows 1 and 2) and those featuring small objects (row 3), we obtained the encoder outputs at the same scales from each model. The feature maps generated by DNOD demonstrate robustness to speckle noise (rows 1 and 2) and capability in representing small objects (row 3). This evaluation underscores the DNOD's representational strength in challenging scenarios.

For visual comparison purposes and to provide insight into the representations learned by our DNOD model, we visualize the feature maps obtained from both the MSFM and DINO encoders, together with the object queries derived from the MADFNO and DINO decoders.

**Encoder Feature Maps:** Two major challenges in SAR object detection are speckle noise and small object size (compared to image resolution). We have visualized our MSFM feature maps in both cases (Figure 6). It is evident that our MSFM operator is robust in tackling both challenges. Furthermore, it ensures a distinct separation between the object and the background, which facilitates the detection ability of the decoder.

**Decoder Query Locations:** We selected object queries with classification scores exceeding 50% and projected these queries onto the feature maps (from encoder) to examine the deformable query locations, thus assessing detection types in each instance (Figure 7). In both instances (top and bottom images), MADFNO decoder, through its utilization of Fourier components and operator properties, is able to target each query on individual objects, using dense representations at the boundaries and on the object's surface (top image). In addition, it accurately outlines boundaries with query locations, aiding in bounding-box detection.

In summary, while DINO's attention is scattered, DNOD effectively concentrates its focus, resulting in sharper boundary localization. From the foregoing discussion, it becomes clear that the integration of our novel neural operators (MSFM and MADFNO) into the DETR framework has markedly enhanced the detection of SAR objects. For additional examples, refer to the Appendix A.10.3.

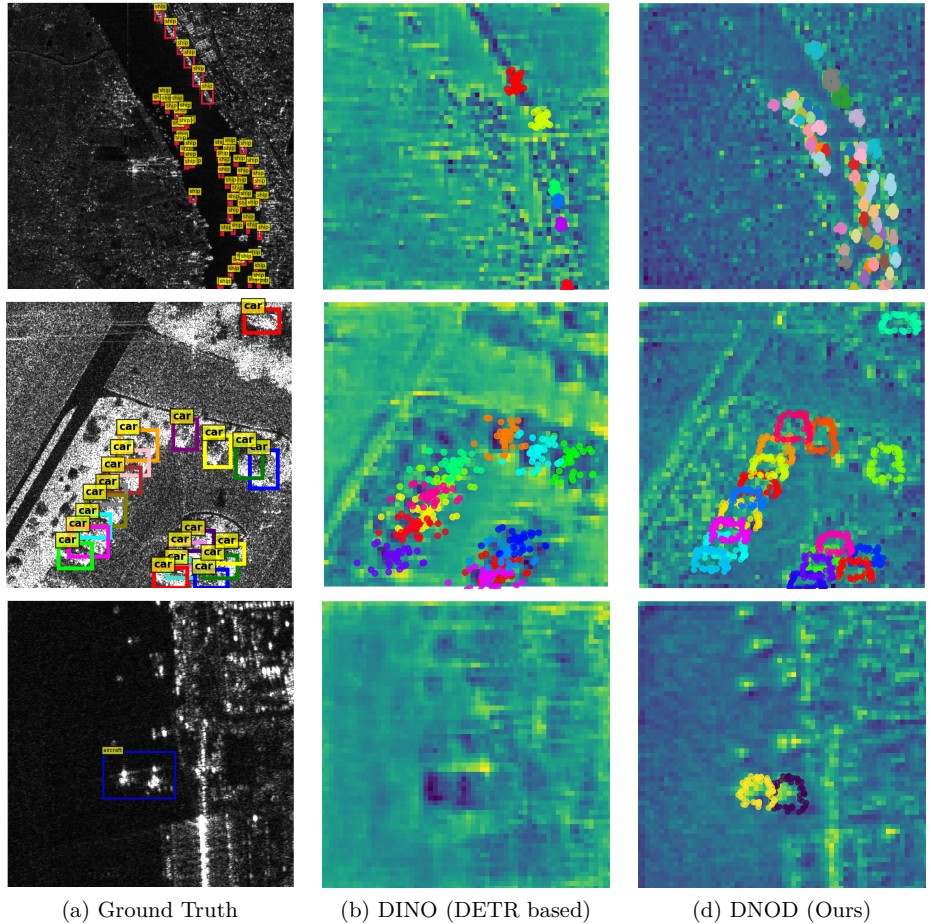

(a) Ground Truth    (b) DINO (DETR based)    (d) DNOD (Ours)

Figure 7: Visualization of feature maps and object queries derived from MSFM–DINO encoders and MADFNO–DINO decoders. MSFM exhibits effective feature extraction and clear object–background separation. MADFNO decoder utilizes Fourier components to densely position queries on object surfaces and edges, facilitating more precise boundary detection than DINO, whose attention remains more dispersed.

## 5  Conclusion and Future Work

We developed DNOD, the first-of-its-kind neural operator-based encoder called MSFM and a decoder called MADFNO within the DETR framework for object detection, showcasing its implementation on SAR datasets. These are new multi-input, multi-scale deformable neural operators. Experimental results and ablation studies show that DNOD offers notable advances over the current leading methods in achieving SoTA performance. Although SAR object detection was the focus here, the utility of our novel architecture with some modifications might have a broader implication for generic object detection and other computer vision tasks using neural operators. Further, SLC/complex phase SAR images would need modifications of both MSFM and MADFNO, which could be attempted in the future. On the whole, the present paper opens exciting new directions for future research.

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

# A  Appendix

## A.1  Related Work

**CNN-based methods:** Convolutional Neural Networks (CNNs) have become particularly successful in computer vision tasks. R-CNN is the breakthrough method that effectively integrated CNNs with region proposals for object detection (Girshick et al., 2014). Advancements include Fast R-CNN (Girshick, 2015), which employs single-stage training with a multi-task loss, and Faster R-CNN (Ren et al., 2015), which integrates the region proposal network for a streamlined end-to-end approach. RetinaNet (Lin et al., 2017) introduced focal loss for effective dense object detection, while (Tian et al., 2019) advanced these approaches using anchor- and proposal-free strategies within a per-pixel framework. Further studies have suggested new training techniques and objectives to improve object detection (Li et al., 2020a; Zhang et al., 2020b; 2021a; Zhu et al., 2020; Zhang et al., 2020b). An alternative line of investigation, as demonstrated by YOLO (Redmon et al., 2016), approaches object detection through a one-step process for the prediction of bounding boxes. Its popularity was driven by its efficiency in real-time applications. Various developments, such as (Chen et al., 2021a) and (Ge et al., 2021), have been built on the YOLO framework.

**DETRs:** With the recent success of transformers (Vaswani et al., 2017) in language modeling, a new paradigm has emerged in object detection, namely DETRs (Carion et al., 2020), which opened new possibilities for integrating encoder-decoder frameworks into object detection tasks. Although this work was not state-of-the-art at the time and suffered from slower convergence problems, it established a new pathway for the field. Subsequently, several works have improved the DETR framework for improved performance and efficiency. Conditional DETR (Meng et al., 2021) introduced a conditional spatial query technique for the decoder, which addressed the convergence problem in DETR. Inspired by deformable convolutions (Dai et al., 2017) in computer vision, Deformable DETR introduced multi-scale deformable attention-based encoders and decoders for improved convergence and spatial resolution. DAB-DETR (Liu et al., 2022a) employed a different query formulation using dynamic anchor boxes. DINO (Zhang et al., 2023a) combined approaches from Zhu et al. (2021) and Liu et al. (2022a), further incorporating denoising queries with a contrastive denoising strategy, achieving superior performance compared to previous models. Various research initiatives, such as Zhao et al. (2024); Lin et al. (2023b); Zang et al. (2022); Li et al. (2023a); Chen et al. (2023); Dai et al. (2021), among others, have proposed several modifications to the initial DETR model.

**SAR Object detection:** In the literature, SAR object detectors are predominantly developed by adapting current state-of-the-art object detection frameworks. Specifically, two-stage approaches, such as Kang et al. (2017), implement modified R-CNN architectures for object detection in SAR imagery. A variety of faster R-CNN adaptations have been presented (Li et al., 2017; 2020b), alongside methodologies derived from RetinaNet (Miao et al., 2022). The Dense Attention Pyramid Networks utilized by DAPN (Cui et al., 2019) facilitated the detection of objects at multiple scales. The LMSD-YOLO framework (Guo et al., 2022) was enhanced with depthwise separable convolutions, batch normalization, and ACON activation functions. YOLO-FA (Zhang et al., 2023b) introduced frequency-adaptive learning components into the YOLO architecture. In line with the advances of DETR in general computer vision, numerous variants of DETR tailored for object detection have emerged in SAR images (Zhang et al., 2024; Feng et al., 2023).

Another direction of research has introduced novel methodologies specifically tailored for object detection in SAR images. Li et al. (2024a) proposed space-frequency selection convolution layers specifically designed for SAR object detection. (Zhang et al., 2025b) proposed a benchmark for rotated SAR object detection. Very recently (Zhang et al., 2025a) have proposed Gamma Distribution PCA enhanced feature learning for SAR target recognition. Li et al. (2024b) developed a Multi-Stage with Filter Augmentation (MSFA) pretraining framework for SAR object detection that adapted existing state-of-the-art methods for SAR applications. DenoDet (Dai et al., 2024) employed a dynamic frequency domain attention module that performs soft thresholding operations in a transformed domain to enhance object detection performance under high speckle noise conditions.

**Neural Operators:** Neural operators (Kovachki et al., 2023) differ from conventional neural networks by learning mappings from functions to functions. Initially proposed for PDE solutions, they have subsequently been applied in computer vision tasks. Super Resolution Neural Operator (SRNO) (Wei & Zhang, 2023)

introduces a neural operator for computer vision tasks. Later, (Guibas et al., 2021) proposed efficient token mixing for transformers to improve vision transformers. Very recently DiffFNO (Liu & Tang, 2025) integrated diffusion models with neural operators and achieved SoTA results in super resolution.

## A.2 DETR framework

DETRs, as initially introduced in (Carion et al., 2020), comprise a CNN backbone for feature extraction, followed by a transformer encoder and decoder (Figure 8). The backbone is typically ResNet-50 (He et al., 2016) pre-trained on ImageNet (Deng et al., 2009). The backbone takes an image $I$ as input and outputs feature representations $F = \text{backbone}(I)$. Positional embeddings are added to these backbone features, and a $1{\times}1$ convolution layer reduces the channel dimension $d$ before feeding into the encoder. The spatial dimensions $H$ and $W$ are flattened to create a $d{\times}HW$ feature map, where $HW$ serves as the sequence length and $d$ as the feature dimension for token mixing in the encoder layer. The encoder outputs refined features $X = \text{encoder}(F + \text{positional embedding})$, which serve as keys and values for the cross-attention mechanism in the decoder.

The decoder receives two inputs: (1) object queries $Q_{\text{init}}$ that serve as queries, and (2) content from the encoder $X$ that provides keys and values. Each decoder layer queries objects within the encoder content to produce final object queries $Q_{\text{final}} = \text{decoder}(X, Q_{\text{init}})$. These object queries are then passed to prediction heads, two fully connected networks (FFNs) that output class probabilities $C$ and bounding boxes $B$, respectively. The entire framework is trained end-to-end using a bipartite matching loss.

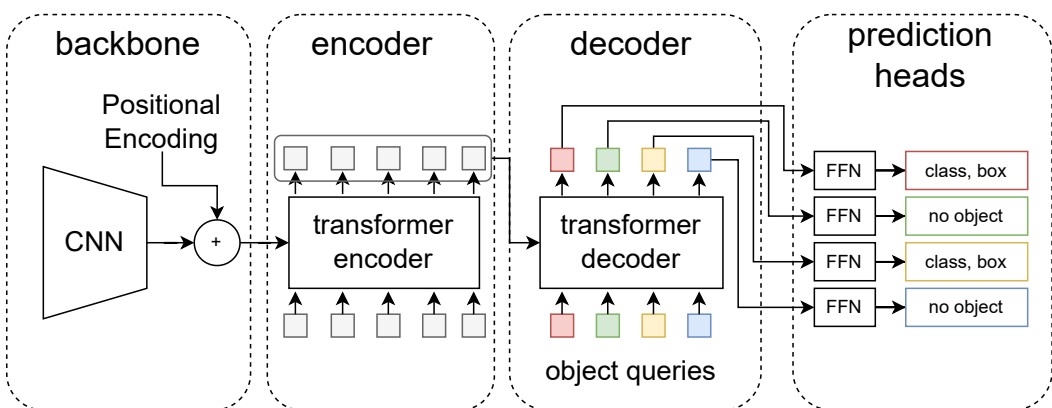

Figure 8: **Overview of the DETR framework:** DETR integrates a CNN backbone with a transformer encoder-decoder to perform object detection. The decoder's attention is directed by position-encoded features from encoder and object queries toward relevant image regions. The final class labels and bounding boxes are obtained through feed-forward networks.

## A.3 Visual examples from SARDet100k dataset

## A.4 Baselines

To evaluate the effectiveness of our model performance, we have conducted a comparative analysis against three distinct categories of object detection models.
*(i)* One-Stage Methods : These methods perform localization and classification in a single pass, i.e directly predict bounding boxes and class probabilities from image pixels. Such as, FCOS (Tian et al., 2019), GFL (Li et al., 2020a), RepPoints (Yang et al., 2019), ATSS (Zhang et al., 2020b), CenterNet (Zhou et al., 2019), PAA (Kim & Lee, 2020), PVT-T Zhou et al., 2022, RetinaNet (Lin et al., 2017), TOOD (Feng et al., 2021), DOOD (Chen et al., 2021b), VFNet (Zhang et al., 2021a), AutoAssign (Zhu et al., 2020), YOLOF (Chen et al., 2021a), YOLOX (Ge et al., 2021).
*(ii)* Two-Stage methods: A sequential pipeline is used in these methods, initially candidate object regions

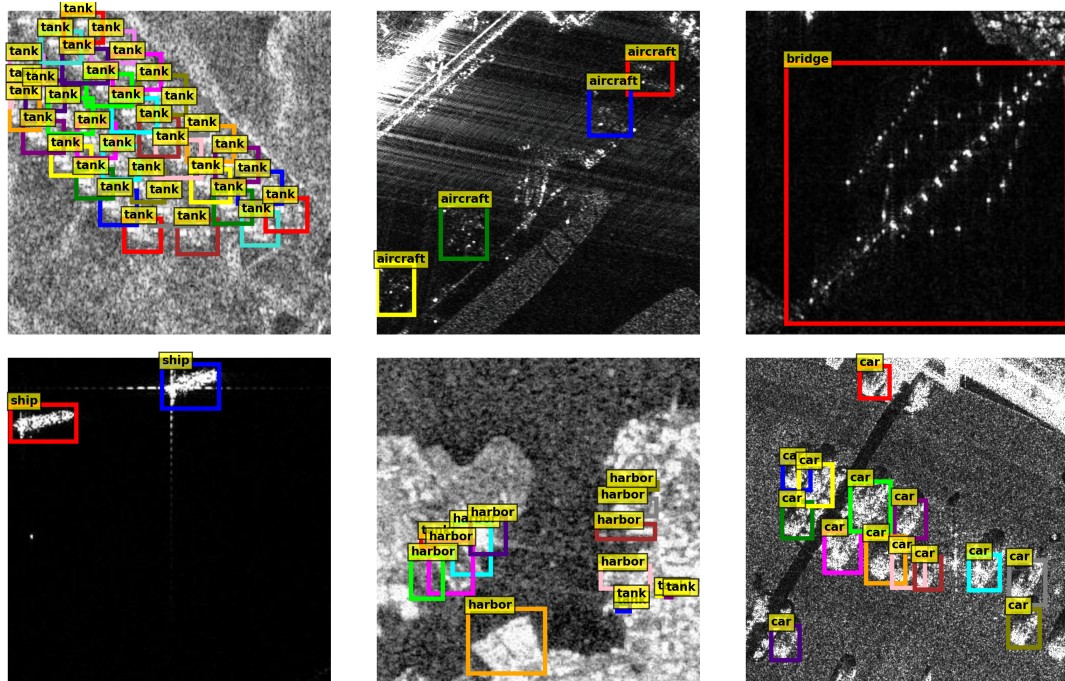

Figure 9: SAR images from the SARDet100k dataset illustrating six object classes: tank, aircraft, bridge, ship, harbor, and car. Each image highlights instances of a specific class using bounding boxes.

are generates using selective search or Region proposal networks. Subsequently, each region is classified and its bounding box refined for accurate object localization. While this approach typically achieves high detection accuracy, it is generally slower than single-stage methods. Such as, Faster R-CNN (Ren et al., 2015), Cascade R-CNN (Cai & Vasconcelos, 2019), Dynamic R-CNN (Zhang et al., 2020a), Grid R-CNN (Lu et al., 2019), Libra R-CNN (Pang et al., 2019), ConvNeXt (Liu et al., 2022b), ConvNeXtV2 (Woo et al., 2023), LSKNet (Li et al., 2023b),

*(iii)* End2End methods: These methods eliminate hand crafted components and uses direct set prediction. Such as, DETR (Carion et al., 2020), Deformable DETR(Zhu et al., 2021), DAB-DETR (Liu et al., 2022a), Conditional DETR (Meng et al., 2021), DenoDet(Dai et al., 2024). This will guarantee a fair, robust, and diverse comparison of our DNOD model for the context of SAR object detection.

## A.5 Pseudocodes for MSFM and MADFNO

This section provides a pseudocode of the proposed operators, MSFM (Figure 10) and MADFNO (Figure 11). The pseudocode methodically describes each step involved, including the fundamental logic, the various inputs, and the specific computations.

## A.6 Hyperaparameters

In Table 5, we present an extensive and thorough compilation of the hyperparameters alongside the training specifications utilized within the construction and application of the DNOD model, providing a comprehensive overview for reference.

```
x = Tensor[b, d, h, w, c]
W_1, W_2 = ComplexTensor[k, c/k, c/k]
b_1, b_2 = ComplexTensor[k, c/k]

def BlockMLP(x):
    x = MatMul(x, W_1) + b_1
    x = ReLU(x)
    return MatMul(x, W_2) + b_2

def MSFM(x):
    bias = x
    x = rfftn(x, dim=(1,2,3))
    x.reshape(b, d, h, w//2 + 1, k, c/k)
    x = BlockMLP(x)
    x.reshape(b, d, h, w//2 + 1, c)
    x = SoftShrink(x)
    x = irfftn(x, dim(1,2,3))
    return x + bias
```

Figure 10: Pseudocode for MSFM with multi scale features, adaptive weight sharing and adaptive masking

Table 5: DNOD Hyperparameter

| Parameter | Value |
|---|---|
| Matcher | HungarianMatcher |
| One to Many matcher threshold | 0.4 |
| One to Many classification loss coefficient | 2 |
| One to Many bounding box loss coefficient | 5 |
| One to Many GIoU loss coefficient | 2 |
| One to One classification loss coefficient | 1 |
| One to One bounding box loss coefficient | 5 |
| One to One GIoU loss coefficient | 2 |
| Positional Embedding type | sine |
| Positional embedding temperature | 20 |
| Number of blocks in Fourier mixing | 8 |
| Focal Alpha | 0.25 |
| Number of classes | 7 |
| Weight Decay | 0.0001 |
| Learning rate | 0.0001 |
| Learning rate drop | 0.1 |
| Hidden dimension | 256 |
| No of deformable decoder points | 6 |
| Non Max Suppression IOU Threshold | 0.8 |
| No of Queries | 1200 |
| Channel Mixing Dimension | 2048 |
| Optimizer | AdamW |

```
def MADFNO(Eo, Q, Rp):
# input -> Eo = Tensor[b, n_l, H, W, c]
#          Q  = Tensor[b, n_q, c]
#          Rp = Tensor[b, n_q, n_l, 4]
# output ->Q_final = Tensor[b, n_q, c]

    bias = Q
    Q = M(Q)
    r = SL(Q)
    r = r.reshape(b, n_q, n_s, n_l, n_p, 2)
    Rp = Rp[:, :, None, :, None, :2]
    T = Rp + r
    Eo = Eo.reshape(b, n_l, H, W, n_s, c//n_s)
    EoS = GridSample(Eo, T) # Bilinear interpolation
    EoS = EoS.reshape(b, n_l*n_p, c)
    z_inp = Concat[EoS, Q, (dim=2)]
    # Shape of z_inp -> (b, n_q, n_l*n_p, c)
    z = rfftn(z_inp, dim=(2))
    z = z.reshape(b, n_q, n_l*n_p//2 + 1, k, c/k)
    z = BlockMLP(z)
    z = z.reshape(b, n_q, n_l*n_p//2 + 1, c)
    z = SoftShrink(z)
    z = irfftn(z, dim(2))
    z = z + z_inp
    Qfinal = z.mean(dim=2)
    return Qfinal + bias

W_1, W_2 = ComplexTensor[k, c/k, c/k]
b_1, b_2 = ComplexTensor[k, c/k]

def BlockMLP(x):
    x = MatMul(x, W_1) + b_1
    x = ReLU(x)
    return MatMul(x, W_2) + b_2

def M(x):
    # input -> x = Tensor[b, n_q, c]
    # output -> x = Tensor[b, n_q, c]
    bias = x
    x = rfftn(x, dim=(1))
    x.reshape(b, n_q//2 + 1, k, c/k)
    x = BlockMLP(x)
    x.reshape(b, n_q//2 + 1, c)
    x = SoftShrink(x)
    x = irfftn(x, dim(1))
    return x + bias
```

```
Eo  -> Encoder
       embeddings
Q   -> Object queries
Rp  -> Reference points
b   -> batch size
n_q -> Number of
       queries
n_l -> Number of levels
n_s -> Number of slices
H   -> Height
W   -> Width
c   -> hidden feature
       dimension
EoS -> Encoder
       embeddings Sampled
Qfinal -> Final Object
       queries
```

Figure 11: Pseudocode for MADFNO with multi-scale features and deformable attention

## A.7    Additional Ablations

### A.7.1    DNOD with Different Backbones

To evaluate the effectiveness of DNOD with other backbones, we experimented with ResNet-18 as a backbone, instead of the ResNet-50 backbone shown in the main paper. The results are tabulated in Table 6. Our

DNOD surpasses DINO and DenoDet, attaining a top mAP of 55.92, with significant enhancements in AP50, AP75, and all object sizes. These findings suggest that the enhanced performance of our DNOD architecture is attributable to the model design, and not to the specific pretrained backbone's architecture.

Table 6: Comparison of SAR Object detection methods on the SARDet-100k dataset with ResNet-18 backbone pretrained on ImageNet.

| Method | Backbone | mAP | AP@50 | AP@75 | $AP_S$ | $AP_M$ | $AP_L$ |
|---|---|---|---|---|---|---|---|
| DINO | ResNet-18 | 44.76 | 78.61 | 46.41 | 36.40 | 56.27 | 55.03 |
| DenoDet | ResNet-18 | 54.36 | 84.40 | 59.12 | 49.10 | 66.61 | 58.01 |
| DNOD (ours) | ResNet-18 | **55.92** | **88.92** | **56.42** | **51.50** | **67.72** | **59.73** |

### A.7.2 DNOD Encoder with Non-NO Spectral Modules

To demonstrate the benefits of neural operators over non-NO frequency or wavelet domain modules, we examined two encoders that utilize frequency or wavelet components along with soft thresholding: the Fourier mixer (Rao et al., 2021) and the wavelet mixer (Patro & Agneeswaran, 2023). These were used in the encoder under the same training conditions and data split, incorporating our Decoder (MADFNO). Table 7 shows the results, illustrating the importance of neural operators in SAR image object detection. Furthermore, our decoder (MADFNO), is the first to use a frequency-based neural operator with cross attention mechanism within the DETR framework for object detection.

Table 7: Comparison of SAR Object detection methods on the **SARDet-100k** dataset with fourier and wavelet mixing modules. Bold indicates it is better than all models. All the models used a ResNet-50 backbone pretrained (Pre.) on ImageNet.

| Method | Backbone | **mAP** | AP@50 | AP@75 | $AP_S$ | $AP_M$ | $AP_L$ |
|---|---|---|---|---|---|---|---|
| DNOD with Fourier Mixers | ResNet-50 | 50.83 | 85.68 | 52.82 | 45.98 | 64.93 | 56.57 |
| DNOD with Wavelet Mixers | ResNet-50 | 39.96 | 72.35 | 39.50 | 34.51 | 50.92 | 40.31 |
| DNOD | ResNet-50 | **56.96** | **90.36** | **59.69** | **52.94** | **71.22** | **65.43** |

### A.7.3 Sensitivity Analysis of DNOD to Scale and Number of Queries

Table 9 presents sensitivity analysis of our DNOD model, evaluating the impact of number of scales (2, 3, 4) and queries (300, 600, 1200). Performance remains largely stable when varying the number of queries from 300 to 1200, with mAP varies within a narrow band of $\pm1.2$ points. The 4-Scale model achieves the overall best results, reaching a peak mAP of 58.11 with 1200 queries, while the 2-Scale model shows slightly lower performance. Increasing the number of queries does not always yield improvements; for example, in the 3-Scale and 2-Scale models, higher queries sometimes lead to minimal drops in AP metrics. These results indicate that DNOD is relatively insensitive to query count, and its detection performance is primarily governed by the scale design rather than the query size.

### A.8 Computational Cost Analysis

We have performed additional experiments for benchmarking GPU-measured latency (ms per image) in terms of Median latency (50th Percentile Latency) and 95th Percentile Latency, Throughput (FPS), Peak Memory (MB), Avg Power (Watt), Energy/Image (Joule) for various batch sizes as shown in Table 10

Figure 12 presents the trade-off between computational complexity (FLOPs) and mean Average Precision (mAP), with circle size representing the number of parameters. Our DNOD variants achieve balance by delivering higher accuracy at significantly lower computational costs than many existing detectors. DNOD and DNOD-large are positioned at the top of the accuracy spectrum, outperforming baselines with comparable or even higher FLOPs. A key observation is that DNOD achieves competitive accuracy while maintaining

Table 8: Accuracy-speed-memory tradeoff for DNOD large for different number of queries and scales.

| Model | Num Queries | mAP | Throughput | Peak Memory |
|---|---|---|---|---|
| **DNOD large 4-Scale** | 300 | 56.88 | 11.76 | 937.10 |
| | 600 | 57.85 | 11.87 | 938.27 |
| | 1200 | 58.11 | 11.22 | 1070.30 |
| **DNOD large 3-Scale** | 300 | 57.01 | 14.02 | 870.96 |
| | 600 | 57.70 | 13.84 | 872.13 |
| | 1200 | 57.51 | 13.00 | 981.06 |
| **DNOD large 2-Scale** | 300 | 56.63 | 16.04 | 804.31 |
| | 600 | 57.00 | 16.46 | 806.45 |
| | 1200 | 56.98 | 16.01 | 889.76 |

Table 9: Performance metrics for DNOD large across different number of queries and scales.

| Model | Num Queries | mAP | AP@50 | AP@75 | $AP_S$ | $AP_M$ | $AP_L$ |
|---|---|---|---|---|---|---|---|
| **DNOD 4-Scale** | 300 | 56.88 | 90.54 | 59.94 | 53.51 | 69.63 | 62.22 |
| | 600 | 57.85 | 91.45 | 60.85 | 55.24 | 70.72 | 63.99 |
| | 1200 | 58.11 | 91.44 | 61.31 | 55.31 | 70.30 | 64.36 |
| **DNOD 3-Scale** | 300 | 57.01 | 90.83 | 60.23 | 53.78 | 69.52 | 64.10 |
| | 600 | 57.70 | 91.00 | 60.89 | 54.97 | 70.33 | 64.39 |
| | 1200 | 57.51 | 91.44 | 60.74 | 55.25 | 70.19 | 64.27 |
| **DNOD 2-Scale** | 300 | 56.63 | 90.27 | 59.21 | 53.34 | 70.02 | 63.76 |
| | 600 | 57.00 | 90.42 | 59.40 | 54.07 | 69.99 | 64.23 |
| | 1200 | 56.98 | 90.82 | 59.95 | 53.57 | 70.57 | 63.13 |

moderate FLOPs, highlighting its efficiency. This analysis demonstrates that DNOD effectively balances performance and efficiency, making it suitable for practical deployment where both accuracy and computational cost are critical.

Table 10: Performance metrics for DINO and DNOD across different batch sizes.

| Model | BS | Median Latency | P95 Latency | Throughput | Peak Memory | Avg. Power | Energy/Image |
|---|---|---|---|---|---|---|---|
| **DINO** | 1 | 28.61 | 29.85 | 34.71 | 332.73 | 192.47 | 5.55 |
| | 2 | 32.85 | 34.02 | 60.54 | 472.15 | 263.91 | 4.36 |
| | 4 | 42.55 | 43.59 | 93.61 | 749.63 | 349.36 | 3.73 |
| | 8 | 75.67 | 76.16 | 105.49 | 1309.63 | 384.55 | 3.65 |
| | 16 | 151.16 | 152.22 | 105.74 | 2425.25 | 386.17 | 3.65 |
| **DNOD** | 1 | 25.41 | 25.84 | 39.22 | 413.90 | 128.10 | 3.27 |
| | 2 | 26.25 | 26.47 | 76.03 | 667.86 | 219.55 | 2.89 |
| | 4 | 33.71 | 33.84 | 118.58 | 1189.54 | 306.99 | 2.59 |
| | 8 | 60.74 | 60.88 | 131.57 | 2229.51 | 332.61 | 2.53 |
| | 16 | 121.78 | 121.90 | 131.36 | 4315.27 | 338.96 | 2.58 |

## A.9 Resolution Invariance

To understand the impact of resolution invariance property of neural operators, Figure 13 visualizes representations extracted from trained models: (a) DNOD (our approach) and (b) DINO. These representations are derived from the same image at different scales. Bilinear interpolation was employed to modify the input scale. Our model demonstrates effective functionality at any resolution, despite having been trained only at a resolution of $512 \times 512$. In contrast, DINO, though specifically designed for multiscale input handling Zhang et al. (2023a), experiences a breakdown beyond certain resolutions. Visualizations also clearly demonstrate that DNOD can distinguish between the background and objects more clearly.

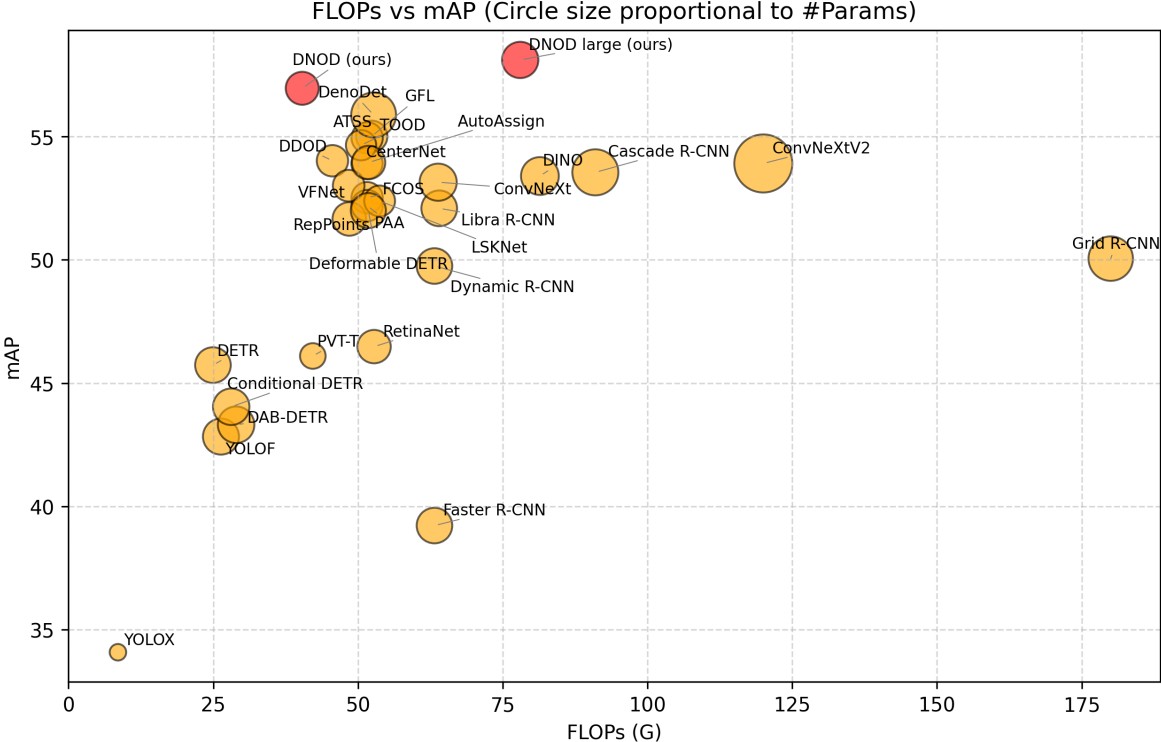

Figure 12: Comparison of FLOPs versus mAP, with circle size proportional to the number of parameters. DNOD achieves high accuracy with moderate computational cost, outperforming several state-of-the-art detectors in efficiency.

DenoDet Dai et al. (2024) is resolution-specific due to the "Deformable Group FC" component in its architecture, which relies on parameters that are resolution dependent. As such, using a different resolution requires retraining with new parameters. Thus, we did not include DenoDet in this experiment. Furthermore, although our framework employs encoder and decoder neural operators, the backbone is limited to ResNet-50, which is not a neural operator. As such, we demonstrate the resolution invariance property solely via visualizations. In future research, we plan to substitute our backbone with a neural operator framework, which allows us to present metrics for various resolution inputs.

## A.10 Additional Results

### A.10.1 Quantitative Analysis Across Object Categories

Table 11 presents an evaluation of our model for each of the six distinct categories (ship, aircraft, car, tank, bridge, harbor) that comprise the SARDet-100k dataset. To determine the model that performs the best in all categories, we implemented a ranking methodology as follows. The models were individually ranked for each category, and then the mean rank was calculated for every model. In particular, our model achieved the lowest mean rank compared to all other models, highlighting its superior performance across all classes.

### A.10.2 Qualitative Analysis in Low Quality Image Context

To augment the qualitative analysis shown in the main paper, we introduced additional comparative studies in low-quality image contexts. Figures 14 and 15 depict a scenario involving the detection of multiple classes (bridge and harbor) in noisy low-resolution SAR images. These figures show that our DNOD accurately detected the bounding boxes for both classes, including multiple instances of the same class.

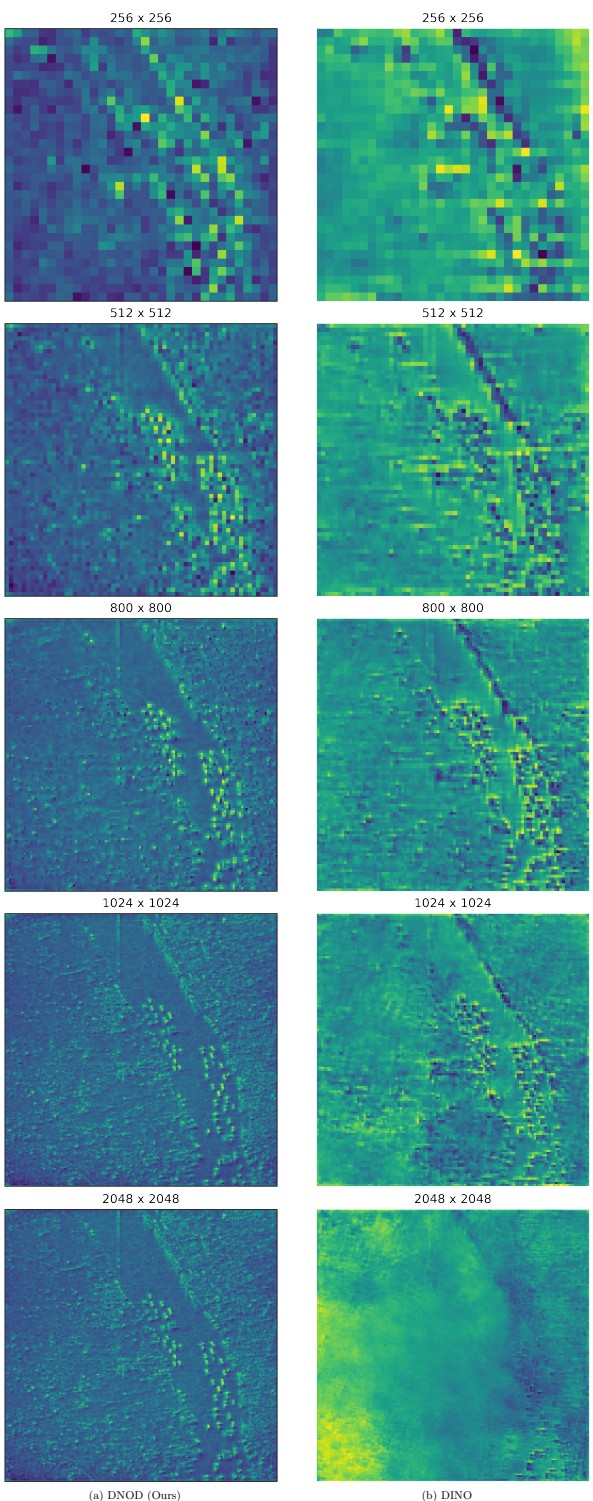

Figure 13: Comparision of (a) DNOD and (b) DINO for resolution invariance by performing inference of pretrained model at multiple scales (256, 512, 800, 1024, 2048)

Table 11: Per-class average precision comparison with SoTA methods on the SARDet-100K dataset.

| Method | Pre. | Ship | Aircraft | Car | Tank | Bridge | Harbor | Avg Rank |
|---|---|---|---|---|---|---|---|---|
| *One-stage* | | | | | | | | |
| FCOS | IN | $59.79_{(21)}$ | $55.44_{(19)}$ | $60.75_{(21)}$ | $41.78_{(11)}$ | $34.17_{(20)}$ | $63.44_{(10)}$ | 17 |
| GFL | IN | $63.92_{(6)}$ | $\mathbf{57.63}_{(1)}$ | $62.29_{(9)}$ | $44.80_{(7)}$ | $36.41_{(9)}$ | $65.04_{(6)}$ | 6.3 |
| RepPoints | IN | $60.85_{(17)}$ | $55.50_{(18)}$ | $61.13_{(19)}$ | $40.69_{(14)}$ | $35.12_{(16)}$ | $56.71_{(22)}$ | 17.6 |
| ATSS | IN | $61.53_{(11)}$ | $55.94_{(12)}$ | $61.77_{(14)}$ | $46.20_{(3)}$ | $37.22_{(5)}$ | $67.48_{(3)}$ | 8.0 |
| CenterNet | IN | $61.24_{(15)}$ | $56.35_{(9)}$ | $61.74_{(15)}$ | $45.31_{(6)}$ | $35.91_{(15)}$ | $63.29_{(11)}$ | 11.8 |
| PAA | IN | $60.16_{(20)}$ | $56.17_{(10)}$ | $60.09_{(22)}$ | $41.07_{(12)}$ | $35.96_{(14)}$ | $60.12_{(17)}$ | 15.8 |
| PVT-T | IN | $53.30_{(25)}$ | $52.91_{(24)}$ | $59.03_{(24)}$ | $30.20_{(24)}$ | $22.51_{(28)}$ | $59.11_{(19)}$ | 24.0 |
| RetinaNet | IN | $55.36_{(23)}$ | $54.00_{(22)}$ | $60.88_{(20)}$ | $32.72_{(23)}$ | $24.81_{(26)}$ | $51.12_{(28)}$ | 23.6 |
| TOOD | IN | $62.28_{(8)}$ | $55.61_{(16)}$ | $62.53_{(7)}$ | $45.96_{(4)}$ | $36.64_{(8)}$ | $65.24_{(5)}$ | 8.0 |
| DDOD | IN | $62.39_{(7)}$ | $56.08_{(11)}$ | $62.48_{(8)}$ | $43.98_{(9)}$ | $36.34_{(11)}$ | $62.87_{(12)}$ | 9.6 |
| VFNet | IN | $62.14_{(9)}$ | $55.84_{(13)}$ | $61.97_{(12)}$ | $42.08_{(10)}$ | $34.11_{(21)}$ | $62.28_{(13)}$ | 13.0 |
| AutoAssign | IN | $62.03_{(10)}$ | $55.70_{(15)}$ | $61.69_{(16)}$ | $\mathbf{48.55}_{(1)}$ | $38.25_{(4)}$ | $57.45_{(21)}$ | 11.6 |
| YOLOF | IN | $52.62_{(28)}$ | $52.64_{(25)}$ | $52.71_{(27)}$ | $22.86_{(29)}$ | $23.74_{(27)}$ | $52.42_{(27)}$ | 27.1 |
| YOLOX | IN | $46.08_{(30)}$ | $46.83_{(30)}$ | $53.43_{(26)}$ | $26.26_{(25)}$ | $13.14_{(30)}$ | $18.95_{(30)}$ | 28.5 |
| *Two-stage* | | | | | | | | |
| Faster R-CNN | IN | $50.45_{(29)}$ | $50.36_{(27)}$ | $57.82_{(25)}$ | $24.90_{(27)}$ | $18.69_{(29)}$ | $33.11_{(29)}$ | 27.6 |
| Cascade R-CNN | IN | $\mathbf{66.99}_{(1)}$ | $56.43_{(8)}$ | $63.25_{(2)}$ | $44.35_{(8)}$ | $36.89_{(6)}$ | $53.81_{(26)}$ | 8.5 |
| Dynamic R-CNN | IN | $61.32_{(13)}$ | $53.86_{(23)}$ | $60.00_{(23)}$ | $33.68_{(22)}$ | $34.40_{(19)}$ | $55.25_{(23)}$ | 20.5 |
| Grid R-CNN | IN | $60.43_{(19)}$ | $55.61_{(16)}$ | $61.94_{(13)}$ | $36.03_{(21)}$ | $31.16_{(23)}$ | $55.13_{(24)}$ | 19.3 |
| Libra R-CNN | IN | $61.32_{(13)}$ | $54.03_{(21)}$ | $61.56_{(17)}$ | $38.12_{(18)}$ | $35.97_{(12)}$ | $61.50_{(14)}$ | 15.8 |
| ConvNeXt | IN | $60.55_{(18)}$ | $57.35_{(3)}$ | $62.13_{(11)}$ | $38.12_{(18)}$ | $36.81_{(7)}$ | $63.95_{(9)}$ | 11.0 |
| ConvNeXtV2 | IN | $61.48_{(12)}$ | $55.83_{(14)}$ | $63.23_{(3)}$ | $39.65_{(16)}$ | $39.16_{(3)}$ | $64.09_{(8)}$ | 9.3 |
| LSKNet | IN | $59.33_{(22)}$ | $56.76_{(6)}$ | $62.74_{(4)}$ | $36.09_{(20)}$ | $35.01_{(17)}$ | $64.38_{(7)}$ | 12.6 |
| *End-to-End* | | | | | | | | |
| DETR | IN | $54.94_{(24)}$ | $51.17_{(26)}$ | $50.11_{(29)}$ | $26.06_{(26)}$ | $32.80_{(22)}$ | $59.31_{(18)}$ | 24.1 |
| Deformable DETR | IN | $60.94_{(16)}$ | $54.16_{(20)}$ | $61.22_{(18)}$ | $39.14_{(17)}$ | $36.09_{(12)}$ | $60.46_{(16)}$ | 16.5 |
| DAB-DETR | IN | $53.16_{(26)}$ | $50.32_{(28)}$ | $49.47_{(30)}$ | $24.06_{(28)}$ | $28.47_{(25)}$ | $55.07_{(25)}$ | 27.0 |
| Conditional DETR | IN | $52.77_{(27)}$ | $49.58_{(29)}$ | $51.00_{(28)}$ | $22.73_{(30)}$ | $29.98_{(24)}$ | $58.16_{(20)}$ | 26.3 |
| DINO 4-Scale | IN | $64.87_{(4)}$ | $56.78_{(5)}$ | $62.72_{(5)}$ | $39.80_{(15)}$ | $34.97_{(18)}$ | $61.26_{(15)}$ | 10.3 |
| DenoDet | IN | $64.91_{(3)}$ | $57.36_{(2)}$ | $\mathbf{63.66}_{(1)}$ | $45.79_{(5)}$ | $36.39_{(10)}$ | $67.17_{(4)}$ | 4.1 |
| ⋆ **DNOD (Ours)** | IN | $64.65_{(5)}$ | $56.48_{(7)}$ | $62.60_{(6)}$ | $40.73_{(13)}$ | $\mathbf{43.27}_{(1)}$ | $\mathbf{74.05}_{(1)}$ | 5.5 |
| ⋆ **DNOD large (Ours)** | IN | $66.31_{(2)}$ | $57.18_{(4)}$ | $62.28_{(10)}$ | $48.40_{(2)}$ | $43.12_{(2)}$ | $71.38_{(2)}$ | **3.6** |

### A.10.3 Additional Visual Insights

Figure 16 displays additional visual representations of DNOD's performance efficacy. Significantly, DNOD accurately localizes bounding boxes compared to DINO, distinguishing it from DETR-based models for object detection in SAR imagery.

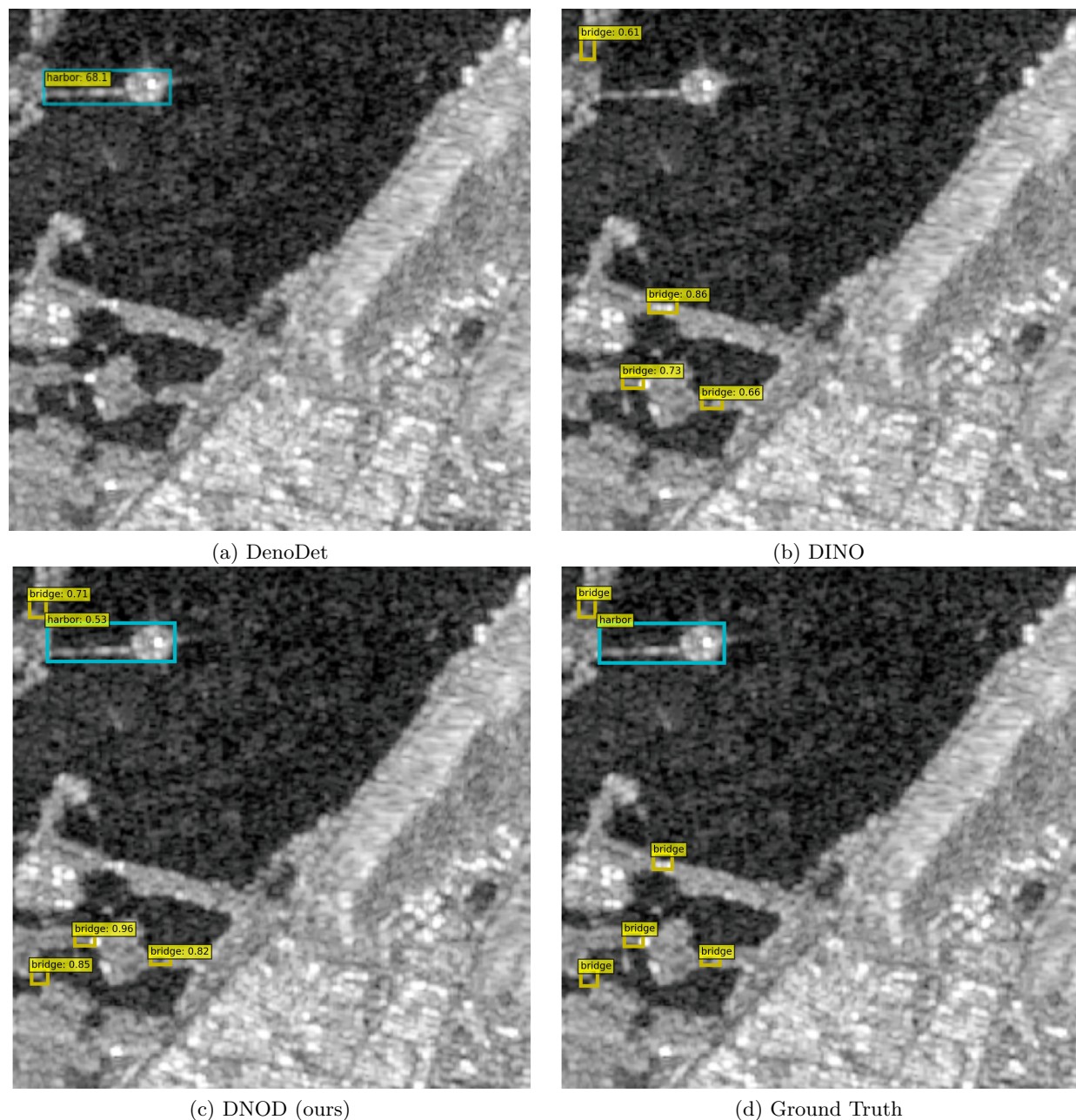

(a) DenoDet

(b) DINO

(c) DNOD (ours)

(d) Ground Truth

Figure 14: Qualitative comparison of DNOD predictions with those of DenoDet (Dai et al., 2024) and DINO (Zhang et al., 2023a) shows that while DenoDet successfully identified only the harbor, and DINO managed to detect only the bridge, our DNOD demonstrated superior performance by accurately identifying both the harbor and the bridge.

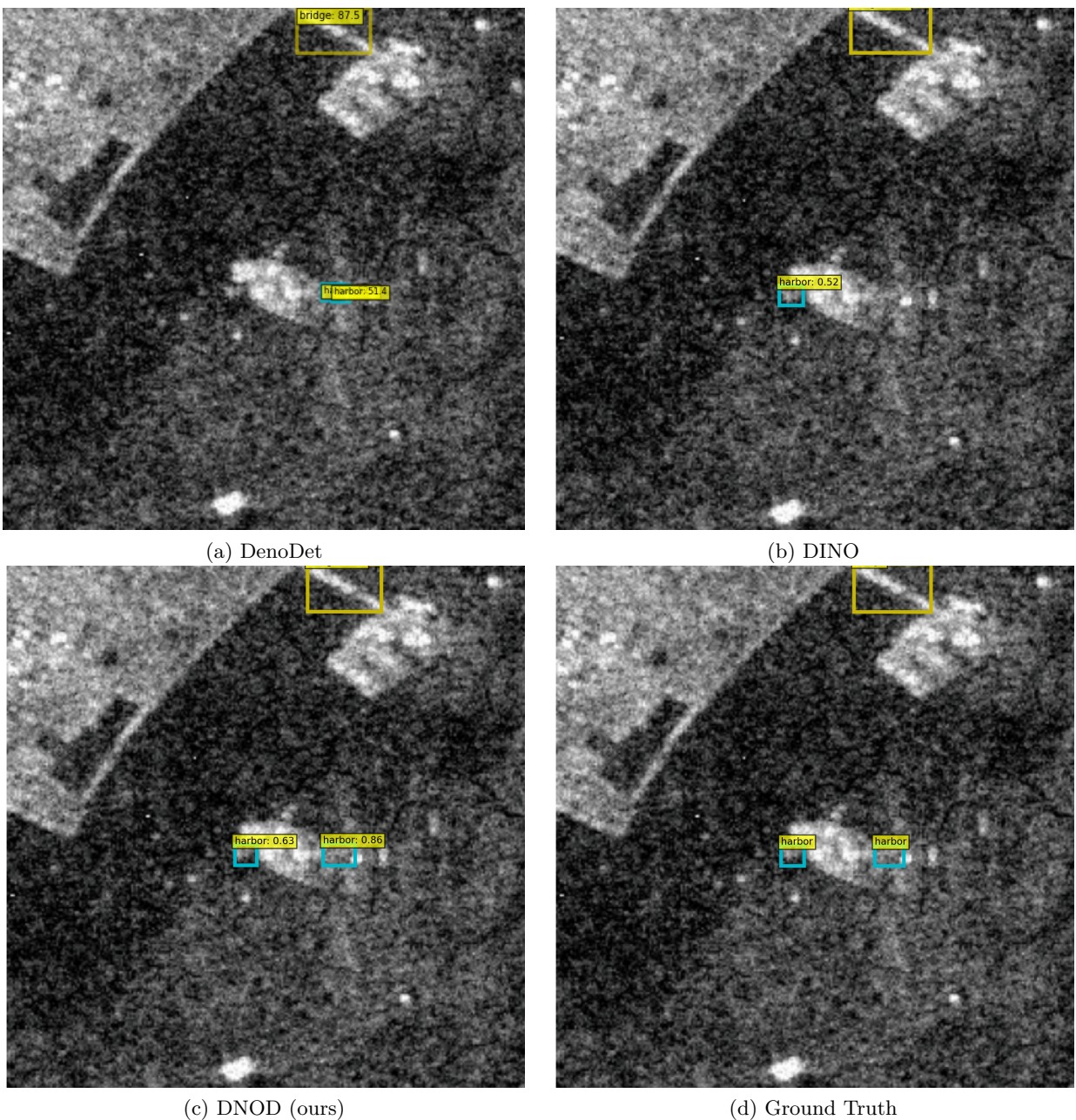

(a) DenoDet           (b) DINO

(c) DNOD (ours)           (d) Ground Truth

Figure 15: Qualitative comparison of DNOD predictions with DenoDet (Dai et al., 2024), and DINO (Zhang et al., 2023a). Both of the baselines only detected a single harbor, but our DNOD detected both the harbors.

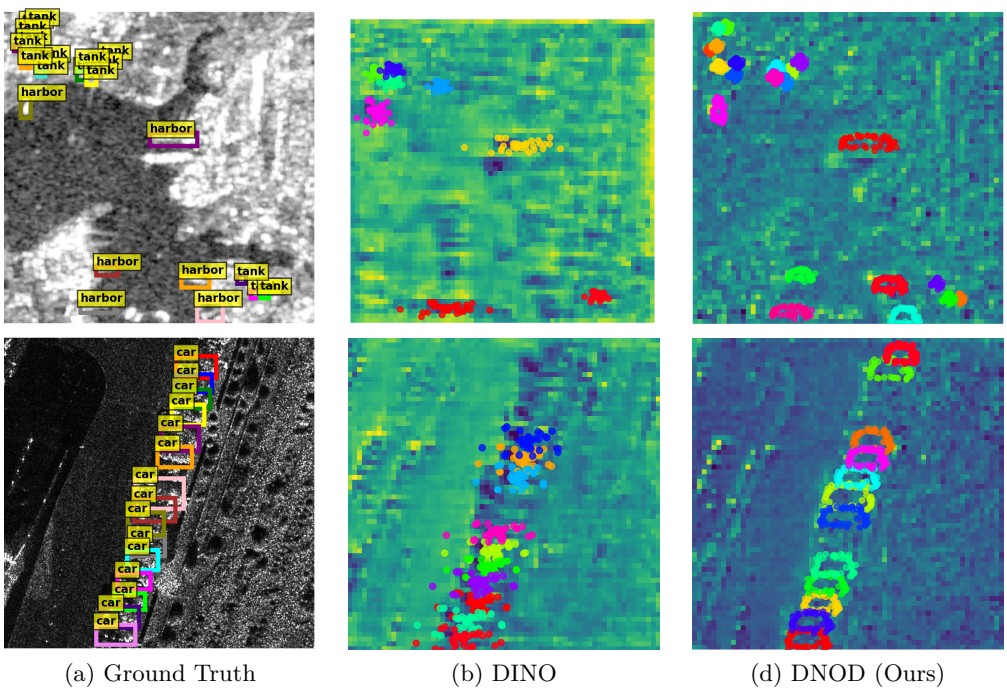

(a) Ground Truth        (b) DINO        (d) DNOD (Ours)

Figure 16: Exploring the Detection Models (a) DINO and (b) DNOD: This is an extension to the analysis from main paper.

