# OpenReview forum: "DNOD: Deformable Neural Operators for Object Detection in SAR Images"
_TMLR — Accepted by TMLR_

### Review · Reviewer_HfNX · 2025-07-17

**Summary Of Contributions:**

This paper integrates two FFT-based operations into a DETR-based model in order to improve on object detection in SAR.  The first (Multi-scale Fourier Mixing "MSFM") applies elementwise multiplication with a kernel learned in the 3d fourier transform space of (scale, x, y), which I believe is basically a 3d conv in the spatial domain with possibly large kernel.  The second (Multi-Scale Adaptive Deformable Fourier Neural Operator "MADFNO") applies a similar learned fourier-space kernel to features sampled from predicted deformable locations (similar to deformable detr, but using fourier mixing of concatenated Q,E features as the mixing operator instead of attention).  The overall system is evaluated on SAR-100K with good improvements compared to existing detectors.

**Additional Comments:**

* what is the input representation to the model?  how is SAR encoded when fed to the model, and is this encoding the same for all comparison baselines?

* it would be helpful to plot a FLOPS vs AP curve for table 2

* eq 6: what is lowercase "m" ?

**Audience:**

Yes

**Audience Explanation:**

This appears to be an effective method for SAR object detection.

**Broader Impact Concerns:**

Since SAR object detection is still somewhat newer than RGB and exclusively applied to satellite data, I think at least mentioning some of the current applications of SAR detection (including surveillance) and the impacts of improved detection on them would be appropriate to help ground the larger context here.  I don't think a separate section is needed for this --- adding this to the Introduction section would be sufficient.

**Claims And Evidence:**

Yes

**Claims Explanation:**

The performance comparisons and ablation with progressive enabling of the components is enough to show improvement over baselines.  Although that may be just enough to answer "yes" here, I think there could be clearer connections between the issues identified (noise vs small objects), and how the mechanics of this method help address them.  See below questions/changes requests.

**Requested Changes:**

While there are mechanical descriptions of the operations the new components perform, there isn't much that illustrates what they do or their behaviors beyond this, or how these particular operations actually address the issues identified with noise.

For example, following the very general description of a fourier kernel on p5, p.6 describes that the component performs "spatial mixing across multiple scales via the Fourier transform ... followed by channel mixing ... The final embeddings are obtained after multiple encoder blocks".  While this states the steps the module does, I would also have liked to see more illustrative examples of what these operations look like, how they behave, and how this is linked to the issues of noise identified before.

Stepping through an example image or region, showing a slice of feature maps and how they behave in the fourier domain vs spatial domain, could help here.  So might measurements on the frequency of the noise somehow compared to the transformations or mixing kernels.  I'm also not sure how this compares to large-kernel depthwise convolutions across space and scale, applied in the spatial domain (does this also help?).

For the second component with deformations, this is even more the case --- I believe the deformable part is done in the spatial domain (though I'm not entirely sure about that) --- but what is the impact of adding fourier mixing to deformations?  Simply showing an improvement is not enough to really see what this does.

---

> ### Author Response · Authors · 2025-09-15
>
> **Q1: While there are mechanical descriptions of the operations the new components perform, there isn't much that illustrates $\cdots$ how this compares to large-kernel depthwise convolutions across space and scale, applied in the spatial domain (does this also help?).**
>
> **A1:** This is an excellent suggestion. Thank you so much. We have now added visual insights on the representations that are learned by our encoder and decoder neural oeprators. A new section 4.4 shows these new visualizations. Furthermore, we have edited the flow of the paper to include a Figure where a visual representation of the encoder outputs and the superior detection ability of our DNOD is clearly highlighted. More such analysis and results are also included in the Appendix 9,10.2,10.3 . We have also added comparision against non neural operator based modules, refer to Appendix 7.2.
>
> **Q2: For example, following the very general description of a fourier kernel on p5, p.6 describes that the component performs "spatial mixing across multiple scales via the Fourier transform $\cdots$ how they behave, and how this is linked to the issues of noise identified before.**
>
> **A2:** Thanks for the comment and suggestion. The pseudocode for the MSFM encoder (and MADFNO decoder) in the Appendix 5 contains the exact steps involved in our encoder (and decoder). These operations contain Fourier kernels and soft thresholding that reduce the effect of speckle noise in SAR images. We also now visually show the encoder representations that clearly demonstrates the effect of our neural operators on images with speckle noise, small objects, and in low resolution settings. For further clarification please refer to Section 4.4 and Appendix 9 and 10.
>
> **Q3: For the second component with deformations, this is even more the case --- I believe the deformable part $\cdots$ Simply showing an improvement is not enough to really see what this does.**
>
> **A3:** Thank you for your insightful comment, we have now visualized the deformable concentration locations and added in section 4.4 in our Main paper (revised version). In summary, while Deformable attention's focus is dispersed, MADFNO pinpoint its concentration effectively, and sharper boundary localization is achieved.
>
> ## Additional comments:
>
> **Q4: What is the input representation to the model? how is SAR encoded when fed to the model, and is this encoding the same for all comparison baselines?**
>
> **A4:** Input image is directly sent into the Resnet-50 Backbone followed by the object detector. It is indeed the same for all the baselines.
>
> **Q5: It would be helpful to plot a FLOPS vs AP curve for table 2**
>
> **A5** We have now added the FLOPS vs AP curve with number of parameters for our model, and all the baselines in our revised version(refer to Figure 12 and Appendix 8).
>
> **Q6: eq 6: what is lowercase "m" ?**
>
> **A6** Thanks for the catch. We have now changed $m$ to $\mu$ to describe it, as it is a kernel function at a specific token s. Please refer to the revised version of paper.

---

### Review · Reviewer_6B9R · 2025-07-27

**Summary Of Contributions:**

The authors propose a new deep learning method, DNOD, for object detection in SAR images, addressing SAR‑specific challenges such as speckle noise and the detection of small objects. In particular, they leverage neural operators by introducing MSFM and MADFNO as the encoder and decoder, respectively, on top of DETR. As a result, they achieve state‑of‑the‑art performance on SARDet‑100K. The contributions of this paper can be summarized as follows:
- First application of neural operators to object detection: This is the first study to introduce neural operator techniques into the object detection domain, especially for SAR imagery, thereby opening a new research direction.
- A novel architecture tailored to SAR: The authors develop two new modules specialized for SAR (the MSFM encoder and the MADFNO decoder) to reduce speckle noise and improve small‑object detection.
- Integration into the DETR framework: They integrate the proposed neural operator modules into a DETR‑based detector and build DNOD, an end‑to‑end trainable and deployable object detection model.
- Demonstration of strong detection performance: On the SARDet‑100K dataset, the method surpasses existing approaches in SOTA accuracy, notably improving mAP and the detection of small and medium objects, while also showing favorable efficiency in terms of accuracy per parameter.

**Additional Comments:**

N/A

**Audience:**

Yes

**Audience Explanation:**

Object detection in SAR images has been studied for a long time, but because the data are satellite images, most targets are small, making the task persistently challenging. How to detect small targets is of high interest to the machine learning community. In addition, unlike general RGB images, SAR has distinctive frequency characteristics and a special sensing modality, so designing methods that handle such data in a principled way is a highly interesting problem.

**Claims And Evidence:**

No

**Claims Explanation:**

The empirical claim of performance gains on SARDet‑100K is relatively well supported from both quantitative and qualitative perspectives. The paper provides comparison tables using the same dataset and evaluation metrics against a large set of baselines (one‑stage, two‑stage, and DETR‑style detectors), showing that DNOD and DNOD‑Large outperform prior best methods (such as DenoDet) in mAP, with especially clear improvements for small and medium objects.

By contrast, the paper’s more central claims, namely "why it has to be a neural operator" and "how discretization invariance alleviates the challenges posed by small objects and scale diversity'', are not directly substantiated by the experiments. The rationale that frequency‑domain mixing and soft‑thresholding for noise suppression are effective is reasonable, but these are not advantages unique to neural operators. Similar effects can be achieved with AFNO‑style token mixers, Fourier/FFT blocks, wavelets, or frequency attention (e.g., DenoDet). There is no head‑to‑head, equal‑recipe comparison under counterfactual conditions ("being an NO" versus "not being an NO but using frequency modules"), nor are NO‑specific consequences directly visualized and quantified (e.g., extrapolation across training/evaluation resolutions, cross‑sensor or cross‑band generalization, superiority under noise‑strength sweeps). Consequently, the necessity claim runs ahead of the presented evidence. Furthermore, because the current evaluation is limited to COCO‑style, amplitude‑only (intensity) SARDet‑100K images and does not incorporate SAR‑intrinsic physical quantities such as phase, coherence, or polarization, the implied advantage of "NOs specifically for SAR" is less convincing.

**Requested Changes:**

- The core claim of this paper is that neural operators (NOs) bring an essential advantage to SAR object detection. To directly support this claim experimentally, please conduct head‑to‑head comparisons against non‑NO frequency‑domain modules (AFNO‑like token mixers, Fourier/FFT blocks, wavelets, DenoDet‑style frequency attention) under the exact same training recipe and data splits. (Critical)

- If you wish to claim ''generalizability to standard object detection,'' please either provide initial experiments on optical datasets such as COCO or, alternatively, tone down the claim by explicitly limiting it to SAR intensity imagery. (Strengthening)

- You state that ``the discretization invariance property of the neural operator will reduce challenges related to small targets.'' However, I did not find experiments that directly demonstrate this effect in the paper. For example, do you have concrete results showing that detection performance is maintained when image resolution is reduced? (Critical)

- Operational metrics for inference speed, memory, and energy: beyond FLOPs and parameter counts, please report GPU‑measured latency (ms per image), throughput (FPS), peak memory usage, and batch‑size constraints. Since your approach makes heavy use of FFTs, provide a clear outlook on real‑time feasibility and include measured comparisons against DenoDet/DINO. (Critical)

- Your evaluation is conducted on intensity (amplitude‑only) images. If feasible, please add small‑scale experiments using SLC/complex phase, coherence, or multi‑polarization (HH/HV/VH/VV) data to demonstrate the value of NOs as complex Fourier operators and as multi‑input operators. If this is not currently feasible, at least include concrete plans and technical prospects for such future extensions in the paper. (Strengthening)

- The number of detectable objects depends on the number of queries. Please present accuracy–speed–memory trade‑off curves when varying the number of queries (e.g., 300/600/900/1200) and the number of scales used (2/3/4). (Critical)

- Please verify in at least one or two settings that the same advantages hold with lighter backbones beyond ResNet‑50 (e.g., ResNet‑18 or Swin‑T). It is important to show that the effect of NOs is not backbone‑dependent. (Critical)

- Augment the exposition with intuitive diagrams and analogies that clarify the relationship between the NO mathematics (kernel integrals, lifting/projection, discretization invariance) and the attention structure in DETR, so that readers outside the immediate subarea can follow the ideas. (Strengthening)

---

> ### Author Response · Authors · 2025-09-15
>
> We thank you for your critical and insightful review. Implementing your suggestions of robust and diverse experiments, we obtained many new findings that have improved our paper.
>
> **Q1: The core claim of this $ \cdots $ please conduct head‑to‑head comparisons against non‑NO frequency‑domain modules (AFNO‑like token mixers, Fourier/FFT blocks, wavelets, DenoDet‑style frequency attention) $ \cdots $  data splits.** $(Critical)$
>
> **A1:** To demonstrate the benefits of neural operators over non-NO frequency or wavelet domain modules, we examined two encoders that utilize frequency and wavelet components along with soft thresholding: the Fourier mixer [1] and the wavelet mixer [2]. These were used in the encoder under the same training conditions and data split, incorporating our Decoder (MADFNO). The table 7  highlights the distinctions between neural operators and non-NO frequency or wavelet domain modules, illustrating the importance of neural operators in SAR images. Notably, we are the first to present a frequency-based neural operator decoder (MADFNO) within the DETR framework. Results are presented in Appendix 7.2.
>
>
> **Q2: If you wish to claim ''generalizability to standard object detection,'' please either provide initial experiments on optical datasets such as COCO or, alternatively, tone down the claim by explicitly limiting it to SAR intensity imagery.** $(Strengthening)$
>
> **A2:** We will tone down the claim by explicitly limiting our results and discussions to SAR intensity imagery, withdrawing any generalizations over standard object detection without experiments on datasets like COCO. We want to consider generalization to other object detection datasets in our future work.
>
> **Q3: You state that ``the discretization invariance property $ \cdots $ is reduced?** $(Critical)$
>
> **A3:** We have added a new section Appendix 9, where we now include new experiments to demonstrate the ability of our model to handle inputs robustly at different scales. Thank you for the suggestion to include these new experiments that strengthen our paper.
>
> **Q4: Operational metrics for inference speed, memory $\cdots$ comparisons against DenoDet/DINO.** $(Critical)$
>
> **A4:** Please refer to the newly added section Appendix 8. In addition to the Parameters and FLOPs in our main paper, we have now performed additional experiments for benchmarking GPU‑measured latency (ms per image) in terms of Median latency (50th Percentile Latency) and 95th Percentile Latency, Throughput (FPS), Peak Memory (MB), Avg Power (Watt), Energy/Image (Joule).
>
> **Q5: Your evaluation is conducted on intensity $\cdots$ such future extensions in the paper.** $(Strengthening)$
>
> **A5:** The dataset we have used already contains Multi-Polar amplitude images (HH/HV/VH/VV) --refer to Table 1. The idea of extending the power of Neural Operators to SLC/complex phase is intriguing, and could be a new line of research in the future. We mention the same in the revised paper.
>
> **Q6: The number of detectable objects $\cdots$ (e.g., 300/600/900/1200) and the number of scales used (2/3/4).** $(Critical)$
>
> **A6:** Thank you for suggestion. We have completed the new experiments and documented the results in Appendix 7.3.
>
> **Q7: Please verify in at least one or two settings that the same advantages $\cdots$ (e.g., ResNet‑18 or Swin‑T). It is important to show that the effect of NOs is not backbone‑dependent.**  $(Critical)$
>
> **A7:** To evaluate the effectiveness of our neural operator based architecture for other backbones, we have considered Resnet-18 as the backbone and trained recent state-of-the-art models. Irrespective of backbone our model demonstarted superior performance as shown in Appendix 7.1.
>
>
> **References:**
>
> [1] Yongming Rao, Wenliang Zhao, Zheng Zhu, Jiwen Lu, and Jie Zhou. Global filter networks for image
> classification. Advances in neural information processing systems (NeurIPS), 34:980–993, 2021.
>
> [2] Badri Patro and Vijay Agneeswaran. Scattering vision transformer: Spectral mixing matters. Advances in
> Neural Information Processing Systems (NeurIPS), 36:54152–54166, 2023.

---

### Review · Reviewer_VMU3 · 2025-09-08

**Summary Of Contributions:**

The authors propose two spectral-type neural network architecture components that are specialized for SAR; these build off the existing DETR architecture. They train/evaluate it on the SARDET-100k dataset, and compare that result with metrics published in the literature.

**Additional Comments:**

I apologize for the tardiness of my response.

**Audience:**

No

**Audience Explanation:**

Maybe I am being too harsh: of course there must be someone in the TMLR audience who want to see the latest SAR numbers. But it seems more like an incremental SAR conference result-- not a machine learning journal result.

**Claims And Evidence:**

No

**Claims Explanation:**

1. The main argument for the paper's novelty is: "this is the first work to introduce neural operators for object detection"
Refutation: "Enhancing Object Detection in Layout Analysis: Leveraging Vision Transformer and Fourier Neural Operator". I don't currently have access but maybe this as well: "Location Invariant Flood Prediction using Fourier Neural Operator"

2. New architecture components are indeed novel

3. Integrating with DETR: this isn't a novel contribution right? It's necessary for your components to function / not a separate thing from #2.

4. You evaluated ONLY the new algorithm; the comparison is copied from the DenoDet paper. That is not a "comprehensive empirical evaluation". Please tell me if I am mis-reading this.

**Requested Changes:**

- Can you justify the new architectural components using principles not specific to SAR? A more detailed results section might be able to speak toward this, if you can characterize what your model is doing differently than the others (what is the nature of the success it found that other models did not)

- Why are the two papers I mentioned above not neural operators applied to object detection?

- "a Fourier transform is executed across scale, height, and width" -- Does this mean Fourier across the scale axis? I've never heard of that: what is the dimension, resolution, and units of this axis?  It looks like you had only "2-4 scales"

- table 2: your bold/underline scheme is inconsistent (dnod regular and large both bold, no underline for AP@50, parameters and flops),

- "The baseline results were obtained from DenoDet (Dai et al., 2024)" -- you should cite Table 2 as a table taken from another paper. It's 93% from that other paper.

---

> ### Author Response · Authors · 2025-09-15
>
> **Are the claims made in the submission supported by accurate, convincing and clear evidence?**
>
> **Q1: The main argument for the paper's novelty is; $\cdots$ Fourier Neural Operator"**
>
> **A1:** *Location Invariant Flood Prediction using Fourier Neural Operator (ICMLA)* : This paper presents an approach  to predict flood events at high spatial resolution, leveraging inundation maps and 15-minute rainfall data. This work is a spatio-temporal prediction or can also be called as 2D time series prediction. There is no object detection in this paper. In the computer vision literature, object detection is generally defined as Classification and Localization of Single or Multiple objects.
>
> *Enhancing Object Detection in Layout Analysis: Leveraging Vision Transformer and Fourier Neural Operator (AINIT)* : The proposed FNO-YOLO model integrates FNO modules into the ViT-YOLO only in the backbone to extract features, demonstrating superior detection performance and reduced computational complexity compared to Self-Attention-based models. This work uses neural operators to enhance feature extraction and subsequent use in a YOLO object detector. Which is not showing any novel capabilities of neural operators for object detection.
>
> **Q2 \& Q3: New architecture components $\cdots$ from \# 2.**
>
> **A2:** We have added new section in the revised version Section 4.4 to discuss the visual reasoning of how integrating these novel neural operator modules have improved the DETRs for SAR object detection. In short, the Multi-Scale Adaptive Deformable Fourier Neural Operator (MADFNO) is focused and concentrated on object boundaries more sharply than Deformable Attention in DETRs for SAR object detection.
>
> **A3:** There are several recent papers in the literature have integrated and modified either encoder or decoder in DETR got accepted in major conferences, which we already discussed in introduction.
>
>
> **Q4: You evaluated ONLY $\cdots$ this.**
>
> **A4:** Indeed, we focused on evaluating our new architecture, and for the baselines, we used the ones reported in the DenoDet paper. For fair evaluation, our DNOD model and the two closest leading models, namely, DINO and DenoDet were re-trained using the same settings as in DenoDet. This ensures direct comparability, and comprehesive evaluation against other baselines reported in DenoDet. We have already mentioned the same in the results section of our paper. Further, we have also cited the DenoDet paper in the table's caption now.
>
> **Would at least some individuals in TMLR's audience be interested in knowing the findings of this paper?**
>
> **Q5: Maybe I am being $\cdots$ journal result.**
>
> **A5:** We reiterate that this is a first neural operator-based object detection paper. Although SAR object detection was the focus here, this will open a new path and will have a broader implication for generic object detection and other computer vision tasks using neural operators, with suitable modifications.
>
> We firmly assert that a considerable portion of the TMLR audience will find object detection tasks relevant, and the innovations we present will prove advantageous for applications that cross- or integrate multiple disciplines. It is important to recognize that detecting small targets in SAR imagery remains a persistent problem in Computer Vision. In addition to size, SAR's distinctive frequency and sensing characteristics require specialized detection techniques. Recently, SAR object detection has received increasing attention as a research focus at prominent AI conferences. For example,
>
> 1) [CVPR - 2025 conference track] Ke Li et al. Unleashing channel potential: Space-frequency selection convolution for SAR object detection. In CVPR, pp. 17323–17332, 2024.
>
> 2) [CVPR - 2025 conference track] Xin Zhang et al. RSAR: Restricted state angle resolver and rotated SAR benchmark. In CVPR, pp. 7416–7426, 2025.
>
> 3) Also, a recent work on SAR target recognition was published in [ICML - 2025 conference track].
> Chong Zhang et al. Gamma distribution PCA-enhanced feature learning for angle-robust SAR target recognition. In ICML, 2025.
>
> 4) The data set used in our paper SARDET-100k is indeed a Spotlight paper from NeurIPS 2024 [conference track].
> Yuxuan Li et al. SARDet-100K: Towards open-source benchmark and toolkit for large-scale SAR object detection. arXiv:2403.06534, 2024.
>
> We have discussed (1,4) in our related work section, and (2,3) are added in the revised version of our paper. In addition, several neural operator papers have recently been published in TMLR such as [1,2,3]
>
> **References:**
>
> [1] Gao, W. et al. Dynamic Schwartz-Fourier neural operator for enhanced expressive power. TMLR 2025.
>
> [2] Kutyniok, G. et al. Probabilistic neural operators for functional uncertainty quantification. TMLR 2025.
>
> [3] Rahman, M.A. et al. U-NO: U-shaped neural operators. TMLR 2024.

---

> > ### Author Response · Authors · 2025-09-15
> >
> > **Requested Changes:**
> >
> > **Q6: Can you justify the new architectural components using principles not specific to SAR? A more detailed results section might be able to speak toward this, if you can characterize what your model is doing differently than the others (what is the nature of the success it found that other models did not)**
> >
> > **A5:** We have added a new section in the revised version of paper Section 4.4 and also added new experiments in Appendix  7-9 to support and more clearly discuss the benefits of our model relative to the baselines.
> >
> > **Q7: Why are the two papers I mentioned above not neural operators applied to object detection?**
> >
> > **A7:** The papers mentioned have not used neural operators for object detection purposes. Refer to A1 for more details.
> >
> > **Q8: "a Fourier transform is executed across scale, height, and width" -- Does this mean Fourier across the scale axis? I've never heard of that: what is the dimension, resolution, and units of this axis? It looks like you had only "2-4 scales"**
> >
> > **A8:** The scale axis that we refer to is 4 scale for DNOD (4 Scale) and 3 Scale for DNOD (3 scale), etc. We have performed fourier mixing along 3 dimensions named height, width and scale, which facilitates inter scale mixing and intra scale mixing. The primary motivation to introduce this novel mechanism is to let the object features from different scale interact with each other to get better refined representations of features, DNOD have indeed achieved these effective representations, please refer to the Figure 6  for representations by our MSFM encoder in our revised paper.
> >
> > **Q9: table 2: your bold/underline scheme is inconsistent (dnod regular and large both bold, no underline for AP@50, parameters and flops)**
> >
> > **A9:** Thank you for your comment. To maintain consistency, we have removed bolding scheme for parameters and FLOPs in our revised version. Note that we have already underlined AP@50 - AutoAssign (One-Stage based baseline) as the second best in that metric.
> >
> > **Q10: "The baseline results were obtained from DenoDet (Dai et al., 2024)" -- you should cite Table 2 as a table taken from another paper. It's 93\% from that other paper.**
> >
> > **A10:** We have already mentioned this in the results section of our paper. For more transparency, we again mention the same in the caption of the Main results table in the revised version of our paper.

---

> > ### Comment · Reviewer_VMU3 · 2025-09-28
> > **Thank you**
> >
> > Thank you for the comprehensive responses. I wanted to call out that I rescind my comment about your results not being a good fit with TMLR-- you are right. Also: Section 4.4's illustration of the Fourier components leading to high frequency representations is a nice touch.

---

### Author Response · Authors · 2025-09-15
**General comment on the revised version.**

- We are sincerely grateful to the reviewers for constructive and insightful suggestions. By following your advice regarding model insights and implementing more rigorous ablations, we have uncovered novel findings that have greatly strengthened our paper.

- All reviewer comments were carefully considered and ensured that every update mentioned in our rebuttal is reflected in the revised paper.

**Below is a summary of the updates made in the revised version:**

1. Added a new section (4.4) titled "Model insights" which describes the internal representations (as depicted in Figure 1, 6, and 7) of the novel neural operators proposed, namely MSFM and MADFNO. This section demonstrates the superiority of our DNOD when compared to other baselines.

2. We have benchmarked DNOD with additional backbone ResNet-18 as shown in Appendix A.7.1. These findings suggest that the enhanced performance of our DNOD architecture is attributable to the model design, and not to the specific pretrained backbone’s architecture.

3. To show the capability of neural operator compared to other Spectral based non neural operator modules we have examined two encoders that utilize frequency and wavelet components as shown in Appendix A.7.2. These results illustrates the importance of neural operators in SAR image object detection.

4. We have performed sensitivity analysis (refer to Appendix A.7.3) of DNOD for different scales and different queries (Table 9) to demonstrate the robustness of proposed model for various queries and scales. Additionally Accuracy-speed-memory tradeoff of our model for different scales and different queries is calculated and documented in Table 8.

5. Additional evaluations in Appendix A.8 contains comprehensive computational cost analysis for benchmarking GPU-measured latency (ms per image) in terms of Median latency (50th Percentile Latency) and 95th Percentile Latency, Throughput (FPS), Peak Memory (MB), Avg Power (Watt), Energy/Image (Joule) for various batch sizes as shown in Table 10. Figure 12
presents comparison of FLOPS vs AP plot along with parameters for DNOD and all the other baselines.

6. Figure 13 illustrates the DNOD's representational power, for different resolutions. (For more details refer to Appendix A.9).

---

### Decision · Action_Editor_VytH · 2025-10-28

**Recommendation:** Accept as is

**Audience:**

Yes

**Audience Explanation:**

The concern from reviewer VMU3 was successfully addressed by the authors' response. Reviewers all find the paper interesting and inspiring, especially in bleeding in domain-specific knowledge into the design of the model.

**Claims And Evidence:**

Yes

**Claims Explanation:**

The authors' response and updates successfully addressed the concerns from reviewers VMU3 and 6B9R. Reviewers find that the additionally added results and visualization helped a lot in supporting the paper's claims.